Methods

# Direct visualization of emergent metastatic features within an ex vivo model of the tumor microenvironment

Libi Anandi*, Jeremy Garcia*, Manon Ros, Libuše Janská, Josephine Liu, Carlos Carmona-Fontaine

**Ischemic conditions such as hypoxia and nutrient starvation, together with interactions with stromal cells, are critical drivers of metastasis. These conditions arise deep within tumor tissues, and thus, observing nascent metastases is exceedingly challenging. We thus developed the 3MIC—an ex vivo model of the tumor microenvironment—to study the emergence of metastatic features in tumor cells in a 3-dimensional (3D) context. Here, tumor cells spontaneously create ischemic-like conditions, allowing us to study how tumor spheroids migrate, invade, and interact with stromal cells under different metabolic conditions. Consistent with previous data, we show that ischemia increases cell migration and invasion, but the 3MIC allowed us to directly observe and perturb cells while they acquire these pro-metastatic features. Interestingly, our results indicate that medium acidification is one of the strongest pro-metastatic cues and also illustrate using the 3MIC to test anti-metastatic drugs on cells experiencing different metabolic conditions. Overall, the 3MIC can help dissecting the complexity of the tumor microenvironment for the direct observation and perturbation of tumor cells during the early metastatic process.**

## Introduction

Most cancer fatalities are directly or indirectly caused by metastases (Siegel et al, 2024). Thus, treating premetastatic tumor cells before they acquire migratory and invasive properties could dramatically reduce cancer mortality (Ganesh & Massagué, 2021). Recent studies using in vivo mouse models have provided novel insights into the metastatic process such as the role played by DNA damage through the promotion of the cGAS/STING pathway (Li et al, 2021), the increased metastatic efficiency of cell groups over single cells (Aceto et al, 2014), key roles of stromal and immune cells (Coffelt et al, 2015; Borriello et al, 2022; Gong et al, 2023), and conditions that facilitate the reactivation of dormant metastases and their engraftment into new tissues (Hu et al, 2023).

The direct observation of nascent metastases remains elusive. Unfortunately, the observation of metastases in vivo requires sophisticated microscopy with costs that are prohibitively expensive for most laboratories (Wyckoff et al, 2000; Nakasone et al, 2012; Bakker et al, 2022). Tools including quantification of circulating tumor cells, histological analyses of biopsies, and warm autopsies (Hoadley et al, 2016; Kumar et al, 2016; Brown et al, 2017; Padmanaban et al, 2019; Gkountela et al, 2019; Iacobuzio-Donahue et al, 2019; Ring et al, 2023) provide important information about late metastatic steps. However, early metastases are much more challenging to detect and to study (Harper et al, 2016; Hosseini et al, 2016; Linde et al, 2018). This challenge is due in part to the complexity of cellular and molecular factors that regulate the emergence of metastasis (Gilkes et al, 2014; Rankin & Giaccia, 2016; Nobre et al, 2018; Massagué & Ganesh, 2021). In addition, the initiation of metastasis is a stochastic process, and thus, when and where a metastatic clone emerges is unpredictable (Chen et al, 2009; Klein, 2009; Makohon-Moore et al, 2018). Amid these limitations, we sought to develop an ex vivo system to model the initial conditions that favor the emergence of metastasis.

A model of a premetastatic tumor must recreate microenvironmental factors that are critical in the regulation of metastasis such as altered metabolic conditions (Gilkes et al, 2014; Rankin & Giaccia, 2016; Nobre et al, 2018; Massagué & Ganesh, 2021), and interactions with immune and stromal cells (Quail & Joyce, 2013; Massagué & Ganesh, 2021). For example, in vivo imaging, molecular, and histological evidence have demonstrated an active role of tumor-infiltrated macrophages (Condeelis & Pollard, 2006; Wyckoff et al, 2007; Wenes et al, 2016; Borriello et al, 2022) and fibroblasts (Gaggioli et al, 2007; Bremnes et al, 2011; Labernadie et al, 2017; Dang et al, 2023) in promoting and facilitating cancer invasion and metastasis. Metabolic stress in the tumor microenvironment is also associated with metastasis promotion (Zhong et al, 1999; Bergers & Benjamin, 2003; Krishnamachary et al, 2003; Robey et al, 2009; Lehúede et al, 2016; Bergers & Fendt, 2021; Lin et al, 2021; Cappellesso et al, 2022). Hypoxia is a common feature of solid tumors, and its pro-metastatic roles are well established (Zhong et al, 1999; Krishnamachary et al, 2003; Wong et al, 2011). However, hypoxia rarely—if ever—occurs alone: as nutrients and oxygen

Center for Genomics & Systems Biology, Department of Biology, New York University, New York, NY, USA

Correspondence: cf97@nyu.edu
*Libi Anandi and Jeremy Garcia contributed equally to this work

diffuse into the tumor mass, oxygen and nutrients in the microenvironment become progressively scarcer, whereas metabolic byproducts, such as lactic acid, accumulate (Thomlinson, 1977; Vaupel et al, 1989; Gatenby & Gillies, 2004, 2008; Gillies & Gatenby, 2007; Lyssiotis & Kimmelman, 2017; Hobson-Gutierrez & Carmona-Fontaine, 2018; Bader et al, 2020). Most likely, multiple conditions within an ischemic microenvironment, including redox stress (Tasdogan et al, 2021), acidosis (Robey et al, 2009; Fais et al, 2014), nutrient starvation (García-Jiménez & Goding, 2019) rather than hypoxia alone, drive the initiation of metastasis.

By definition, ischemic conditions usually arise deep within tumors, and thus, accessing and observing emergent metastases in vivo have been virtually impossible. Some aspects of tumor biology are well captured by ex vivo cultures such as organoids (Gao et al, 2014; Boj et al, 2015; Neal et al, 2018; Vlachogiannis et al, 2018; Kopper et al, 2019; Tuveson & Clevers, 2019; Jacob et al, 2020; Guillen et al, 2022) and other 3D tumor model systems (Edmondson et al, 2014; Anguiano et al, 2017, 2020; Cavo et al, 2018). Still, in these models, ischemic tumor cells remain buried within these structures, and thus, imaging tumor–stroma interactions in those regions poses an almost insurmountable challenge.

Here, we present the development of an ex vivo model of the tumor microenvironment designed to visualize the transition of poorly motile primary tumor cells into migratory metastatic-like cells. This 3D microenvironment chamber (3MIC) models key tumor features including the infiltration of immune cells and the spontaneous formation of metabolic gradients that mimic the metabolic conditions within tumors. Because of its unique geometry, the 3MIC easily allows imaging ischemic cells with unprecedented temporal and spatial resolution. Using this system, we show that ischemic-like environments directly drive the emergence of metastatic features including increased cell migration, degradation of the extracellular matrix (ECM), and loss of epithelial features. Combining in vivo experiments with 3MIC cultures, we showed that these changes were reversible, suggesting that metastatic features can arise even in the absence of clonal selection by hypoxia or other environmental challenges. We also show that tumor interactions with stromal cells such as macrophages and endothelial cells increase the pro-metastatic effects of ischemia. Finally, we illustrate how the 3MIC can be used to test how local metabolic conditions may affect drug responses. In all, the 3MIC is an affordable cell culture system where different components of the tumor microenvironment can be carefully dissected. Its amenability for live imaging can complement in vivo studies to better understand and treat the emergence of metastases.

# Results

### A 3D ex vivo model of the tumor microenvironment for the direct visualization of ischemic cells

Insufficient vascularization and excessive cell growth leave large ischemic regions within solid tumors (Thomlinson, 1977; Vaupel et al, 1989; Gatenby & Gillies, 2004, 2008; Gillies & Gatenby, 2007; Lyssiotis & Kimmelman, 2017; Hobson-Gutierrez & Carmona-Fontaine, 2018; Bader et al, 2020). Hypoxia, acidosis, and other metabolic stressors associated with ischemia are known drivers of metastasis. Because these metabolic conditions often arise deep within tumors, visualizing the emergence of metastatic properties in vivo or in 3D organoids presents unique technical challenges (Fig 1A). To overcome these limitations, we first thought to use the **me**tabolic **mi**croenvironment **c**hamber (MEMIC), an ex vivo model of the tumor microenvironment that we developed to study the impact of ischemic stress on cell cultures (Carmona-Fontaine et al, 2013, 2017; Janská et al, 2021). In this system, cell monolayers form reproducible gradients of ischemia, and by design, imaging ischemic cells is as easy as imaging well-nurtured cells.

Key features of metastatic cells involve morphological changes that are better modeled by 3D tumor structures than by cell monolayers. The MEMIC was designed for 2D cultures, and the density of these cells is crucial for the formation of metabolic gradients (Janská et al, 2021). Unfortunately, this system did not allow achieving enough cell density in 3D cultures to form detectable ischemic gradients (Fig S1A). We thus designed and developed a new system—which we named **3**D **mi**croenvironment **c**hamber (3MIC)—to study the impacts of the metabolic microenvironment on 3D cultures. The principle behind the 3MIC is similar to that of the MEMIC: we grow a dense monolayer of cells inside a small chamber that is restricted from accessing nutrients and oxygen from all sides but one. The opening on this side connects to a large volume of fresh media, and thus, it acts as a source of nutrients and oxygen, whereas cells inside the chamber act as resource sinks. In the 3MIC, however, these cells are not the focus of our analyses and only act as nutrient consumers. These "consumer cells" grow upside down on a coverslip at the top of the chamber, and underneath them, we can introduce 3D tumor structures embedded in the ECM. This two-tier system should allow us to have a high-density consumer cell layer producing strong metabolic gradients within a microenvironment shared by much sparser 3D tumor structures that in turn have a negligible impact on the metabolic conditions of the chamber (Figs 1B and S1B). The location of consumer cells on the top of the chamber allows an unobstructed view of the 3D tumor structures, which will be closer to the objective of an inverted microscope (Fig 1B). These experiments can use different combinations of cell types to form spheroids and the consumer layer. For simplicity, however, and unless mentioned otherwise, experiments shown here use the same cell type for both roles.

To test the 3MIC, we first confirmed that consumer cells form metabolic gradients. As a proxy for resource deprivation, we used a fluorescent hypoxia sensor (HRE-GFP [Janská et al, 2021]) which we expressed in a panel of epithelial tumor cells co-expressing constitutive plasma membrane fluorescence (mCherry-CAAX [Janská et al, 2021]). As in our MEMIC experiments, 2D monolayers of consumer cells formed strong ischemic gradients. For example, spheroids formed by Lung KP cells (derived from a murine model of lung adenocarcinoma driven by $KRAS^{G12D}$ and loss of TP53 [Sánchez-Rivera et al, 2014]) displayed a strong gradient of GFP that increased with the distance to the opening of the 3MIC (Fig S1C). We then aggregated cells into small tumor spheroids that we embedded in a collagen- and laminin-rich ECM scaffold and introduced them into 3MICs with or without monolayers of consumer cells.

**A**

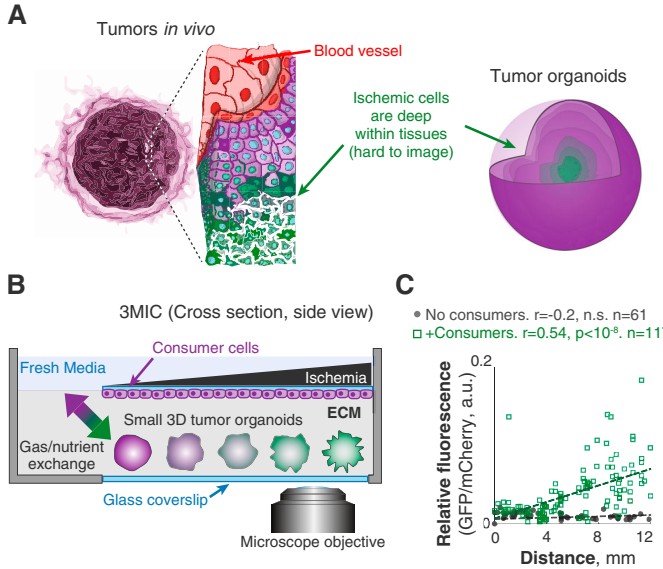

**B**

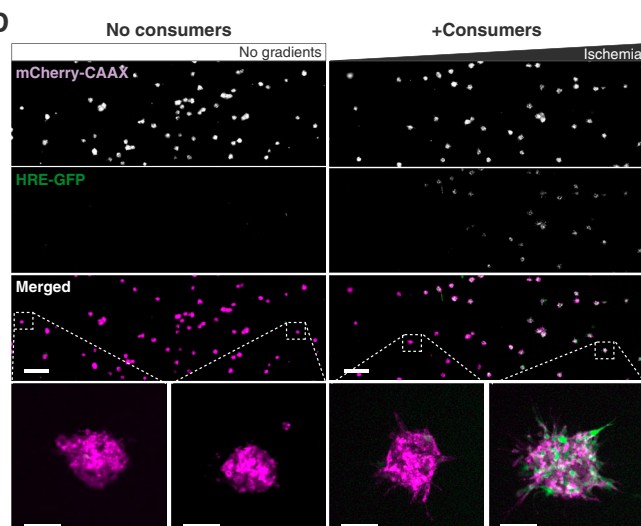

**Figure 1. Design and implementation of a metabolic microenvironment chamber for 3D cultures.**
**(A)** Sketch showing ischemic regions in solid tumor and conventional tumor organoids. Cells in these regions are hard to access and to image. **(B)** Schematic cross-section of the 3MIC where a monolayer of "consumer" cells grows on the top of a small chamber. Tumor spheroids grow embedded in the ECM within the same chamber and underneath consumer cells. This small chamber is connected to a large reservoir of fresh media from only one side (left) where nutrient and gas exchanges occur. **(C)** Quantification of *per* spheroid GFP levels from HRE-GFP Lung KP 3D spheroids cultured in the 3MIC (~100 cells/spheroid) with or without consumer cells. GFP levels serve as a proxy for hypoxia and are normalized to constitutive mCherry-CAAX signal. Dashed line: linear fit. Data are from a representative example of four biological replicates. **(D)** Low magnification showing GFP and mCherry fluorescences in tumor spheroids cultured alone (left column) or in the presence of consumer cells (right column) along the 3MIC gradient. Bottom panels zoom into representative individual tumor spheroids. Bars: 1,000 and 100 $\mu m$ for insets.

As expected, spheroids in control 3MICs without consumer cells did not show signs of hypoxia (Fig 1C and D). In contrast, tumor spheroids growing in 3MICs under consumer cells showed levels of the GFP hypoxia reporter that increased with the distance to the opening of the chamber (Fig 1C and D). We obtained similar results with three independently derived clones of Lung KP cells and in experiments using different cell lines, and in different cell types expressing a different hypoxia reporter (Serganova et al, 2011) (Fig S1D). These results show that consumer cells in the 3MIC form metabolic gradients that influence the local metabolic microenvironment of neighboring tumor spheroids. As these metabolic gradients in the 3MIC are perpendicular to our imaging path, we can observe how ischemic conditions affect tumor structures with unprecedented spatial and temporal resolution.

## Hypoxia is required but not sufficient to increase invasive features in tumor spheroids

In these experiments, we observed that the morphology of tumor spheroids changed dramatically along the 3MIC. Ischemic spheroids acquired a ruffled and protrusive morphology suggesting an increased invasive behavior (Fig 2A; Video 1). In contrast, well-nurtured spheroids close to the opening had a modest size increase and remained mostly smooth and rounded (Fig 2A; Video 1). We decided to take advantage of the unique features of the 3MIC to identify the specific conditions that trigger these seemingly pro-metastatic features on ischemic spheroids.

There is ample evidence linking hypoxia to pro-metastatic tumor features (Zhong et al, 1999; Bergers & Benjamin, 2003; Krishnamachary et al, 2003; Robey et al, 2009; Lehúede et al, 2016; Bergers & Fendt, 2021; Lin et al, 2021; Cappellesso et al, 2022). However, we examined whether additional metabolic stressors—such as nutrient deprivation or lactic acid accumulation—may contribute to the morphological changes we observe in the 3MIC. To rapidly compare the effects of different treatments on spheroid morphology, we defined the *invasiveness index* as the inverse of their circularity. This index increases in cellular structures with irregular and protrusive shapes and remains small in smooth and homogeneous tumor spheroids (Fig 2B).

We first tested whether hypoxia was sufficient to trigger morphological changes and increase cell migration in tumor spheroids. We compared spheroids cultured in regular petri dishes incubated under different oxygen tensions (21% and 1%) with spheroids cultured in the 3MIC. Tumor spheroids cultured in a hypoxic incubator increased levels of the GFP-based hypoxia sensor to levels comparable to ischemic clusters in the 3MIC (Fig S2A). Hypoxic spheroids, however, were not as invasive as spheroids in the 3MIC and were only slightly more invasive than spheroids cultured in regular incubators (Fig 2C). In similar experiments, we treated spheroids with dimethyloxalylglycine (DMOG) or cobalt chloride (CoCl$_2$)—two prolyl hydroxylation inhibitors that stabilize the transcription factor hypoxia-inducible factor 1-alpha (HIF1A) and trigger a hypoxia-like transcriptional response under normal oxygen tension (Chan et al, 2002). As shown in Fig 2D, these spheroids showed only modest increases in invasion despite the strong activation of a HIF1A response, as denoted by the strong increase in the GFP signal from the hypoxia reporter (Figs 2D and S2B). These results show that hypoxia alone triggers relatively minor morphological changes and cannot fully recapitulate the changes that we observe in ischemic regions of the 3MIC.

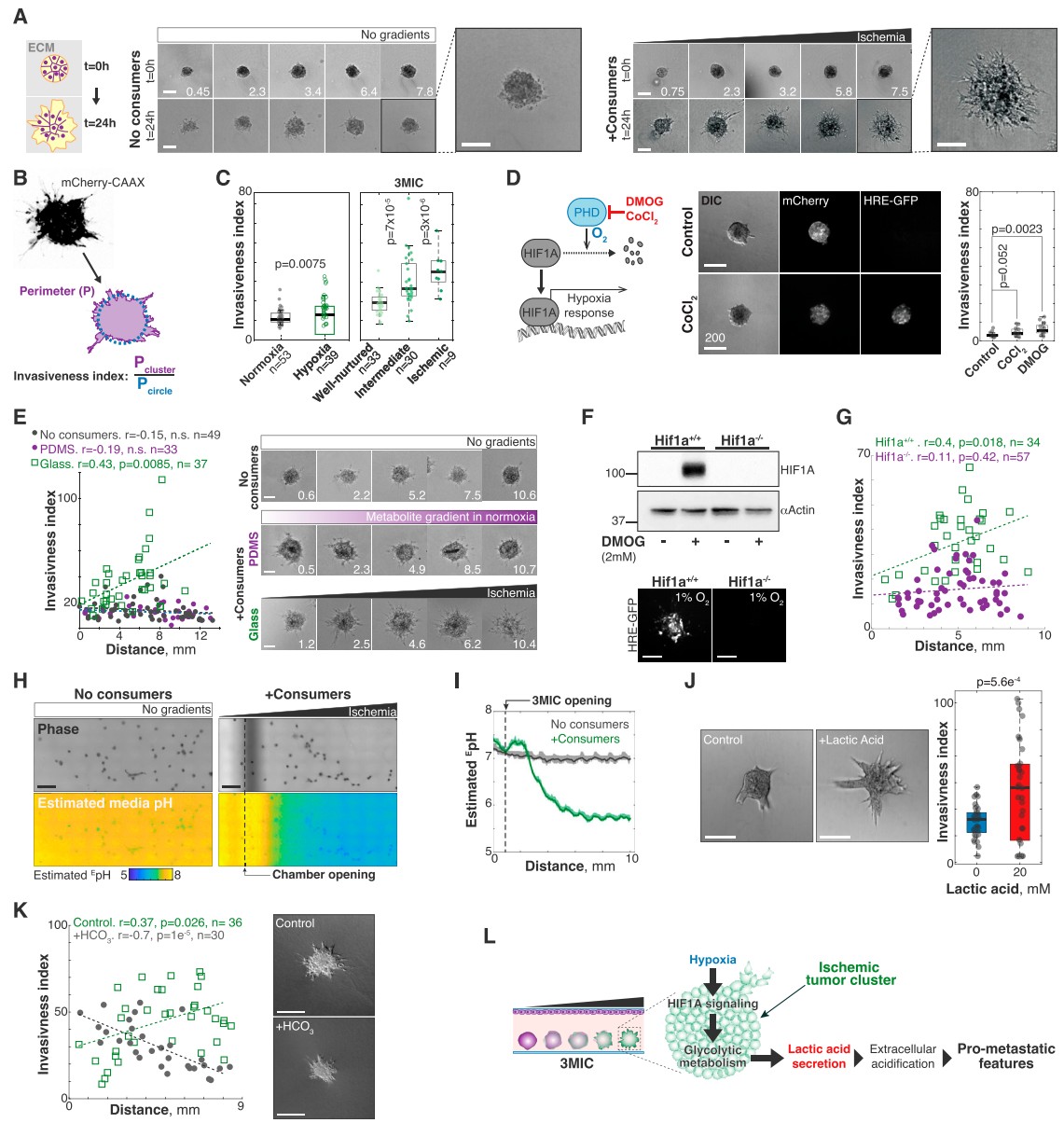

**Figure 2. Low pH is responsible for the increased invasion of ischemic clusters in a HIF1A-dependent manner.**

**(A)** Left: Lung KP spheroids imaged after 0 h (top) and 24 h (bottom) of growth in different regions of the 3MIC without consumer cells. These clusters grew uniformly and retained smooth and round shapes. Right: similar data but for spheroids grown in the presence of consumer cells. Ischemic spheroids grew larger and more protrusively. Numbers in the lower right corner denote the distance to the opening in mm. Bars: 100 $\mu$m. **(B)** We defined the invasiveness index as the perimeter of tumor spheroid over the perimeter of a circle with the same area (1/circularity). **(C)** Using the invasiveness index, we quantified the invasiveness of Lung KP spheroids under different environments. Well-nurtured, intermediate, and ischemic conditions correspond to different regions of the 3MIC. Data points: *per* spheroid invasiveness. Data are from a representative example of two biological replicates. **(D)** Representative images and the invasiveness index of untreated tumor spheroids compared to spheroids treated with $CoCl_2$ or DMOG. Bars: 200 $\mu$m. See Fig S2 for images of DMOG-treated spheroids. **(E)** Representative images of individual spheroids grown for 24 h in the absence of consumer cells (top), or in the presence of consumer cells in 3MICs enclosed in polydimethylsiloxane or glass (center and bottom, respectively). Numbers in the lower right corner denote the distance to the chamber's opening in mm. Bars: 100 $\mu$m. Plots on the right show a quantification of these changes. Data points: the invasiveness index of individual spheroids. Dashed lines: linear fit. Data are from a representative experiment of two biological replicates. **(F)** Top: anti-HIF1A blots of Hif1a$^{+/+}$ and Hif1a$^{-/-}$ cells treated or untreated with DMOG. Bottom: representative images of Hif1a$^{+/+}$ and Hif1a$^{-/-}$ HRE-GFP tumor spheroids cultured in a hypoxic incubator. **(G)** Quantification of the invasiveness index of Hif1a$^{+/+}$ and Hif1a$^{-/-}$ spheroids along the 3MIC. Bars: 200 $\mu$m. **(H)** Representative ratiometric images used to estimate the extracellular pH ($^{E}$pH) within 3MICs with or without consumer cells. **(I)** Estimated extracellular pH over distance from the opening in 3MICs with or without consumer cells. **(J)** Representative images and quantification of the invasiveness index of Lung KP spheroids treated with 20 mM of lactic acid. Bars: 200 $\mu$m. **(K)** Representative images and quantification of the invasiveness index of Lung KP spheroids within 3MICs treated with 25 mM of bicarbonate. Bars: 200 $\mu$m. **(L)** Model suggested by our data where increased invasion under ischemia is driven by low extracellular pH through a HIF1A-dependent mechanism.

Additional experiments, however, revealed that although not *sufficient*, hypoxia was *required* in altering spheroid morphologies in ischemia. First, we took advantage of the selective permeability of polydimethylsiloxane (PDMS) (Carmona-Fontaine et al, 2013, 2017; Janská et al, 2021). PDMS is permeable to oxygen and other gases, but not to soluble metabolites (Cox & Dunn, 1986). We thus constructed 3MICs using a PDMS membrane molded into the shape of a glass coverslip. In these modified 3MICs, cells form gradients of nutrients and metabolic by-products, but oxygen rapidly equilibrates across the PDMS membrane, and thus, there is no hypoxia (Fig S2C). Under these conditions, tumor spheroids did not show major morphological challenges, regardless of whether consumer cells were present or not (Figs 2E and S2C). Second, we established that the changes displayed by ischemic spheroids require HIF1A signaling. Using CRISPR/Cas9, we generated isogenic knockout Lung KP cells (Hif1a$^{-/-}$). As expected, Hif1a$^{-/-}$ cells did not have detectable levels of HIF1A—even after treatment with DMOG (Fig 2F)—and did not increase their hypoxia-driven GFP levels when cultured in a hypoxic incubator (Fig S2D). As in previous experiments, isogenic WT cells (Hif1a$^{+/+}$) showed strong morphological changes along the 3MIC, but these changes were severely impaired in spheroids formed by Hif1a$^{-/-}$ cells (Fig 2G). Altogether, these results show that hypoxia and HIF1A signaling are required but not sufficient for the morphological changes seen in ischemic clusters. As the effects of these perturbations alone are weak, we thought that cells need additional metabolic cues to fully adopt the morphologies we observe in ischemic spheroids.

## Medium acidification is critical for the increase in metastatic features displayed by ischemic tumor cells

We then sought for additional factors that may increase the invasive capacity of tumor spheroids. A simple look with the naked eye at a 3MIC culture would suggest that strong changes in pH are occurring inside. We regularly use cell culture media with the pH indicator phenol red, which shows its typical red color near the opening of the chamber gradually yellowing deeper with the 3MIC (Fig S2E). A ratiometric analysis of this change showed that extracellular pH levels drastically decrease in ischemic regions, whereas they remained relatively constant in 3MICs without consumer cells (Fig 2H and I).

Changes in intracellular pH were much milder (Fig S2F). We thus focused on factors affecting medium acidification such as lactic acid, which is abundantly produced and secreted by the glycolytic metabolism of tumors. Surprisingly, the simple addition of lactic acid to tumor spheroids under otherwise well-perfused and normal conditions strongly increased their invasiveness (Fig 2J). Consistent with a key role of low pH in promoting cell invasion, the addition of bicarbonate to increase the buffering capacity of the media eliminated the invasiveness of ischemic spheroids in the 3MIC (Fig 2K). Importantly, acidification of the extracellular environment has been linked to increased tumor invasion and metastasis (Martínez-Zaguilán et al, 1996; Robey et al, 2009; Cappellesso et al, 2022).

Together, these results show that HIF1A signaling is required to increase the invasion of ischemic spheroids, whereas a low pH is sufficient to do so. We speculate that these effects are linked as HIF1A increases lactic acid secretion through its regulation of

glycolysis (Lum et al, 2007). We think that in normal cultures, or in well-perfused tissues, lactic acid levels do not rise enough to produce a strong pH decrease. But in the 3MIC or in poorly vascularized tissues such as solid tumors, lactic acid is not cleared rapidly enough, and thus, it accumulates increasing cell invasion through a pH reduction (Fig 2L). These experiments illustrate the importance of considering ischemia as a whole rather than hypoxia alone, and they illustrate the ease of modeling these complex metabolic microenvironments in the 3MIC.

## Ischemia stimulates persistent cell migration

The increased invasion of ischemic spheroids was conserved across a panel of human and mouse cell lines (Fig S3A and B). To better understand the mechanisms behind these changes, we first ruled out a major contribution to cell proliferation changes that can also drive the expansion of cell populations (Trepat et al, 2009; Deforet et al, 2019). In fact, only a small number of cells in our 3D cultures were actively proliferating as revealed by a sparse signal for phospho-H3 immunostaining (Fig S3C) and by FUCCI-based live imaging of the cell cycle (Fig S3D; Video 2).

In contrast, live imaging analyses showed a significantly higher cell dispersion in ischemic clusters quantified as a decrease in local cell density (Fig 3A and B; Video 3) (Carmona-Fontaine et al, 2013). Taking advantage of the high temporal and spatial resolution in live microscopy afforded by the 3MIC, we measured the migratory properties of individual tumor cells. Cells in these 3D structures are tightly packed muddling cell detection and tracking. To avoid this problem, we formed chimeric spheroids by mixing two populations of Lung KP cells expressing a nuclear fluorescent protein (H2B-YFP) or mCherry-CAAX typically in a 1:10 ratio (Fig 3C; Video 4). We can then easily track H2B-YFP–positive cells as they are much sparser, and their nuclear label is easy to detect. Using automated image analysis tools, we tracked hundreds of cells from different spheroids along the gradient of ischemia. Analysis of nearly 900 tracks of individual cells showed large heterogeneity in their motility where some cells wandered near their origins, whereas others moved over large distances. On average, however, cells from ischemic clusters traveled further from their original location than cells from well-nurtured clusters (Figs 3D and S3E and F; Video 4). This increase in net displacement was not due to an increase in speed (Fig S3G) but rather because they moved more persistently than well-nurtured cells (Fig 3E). Consistent with their increased persistence, the movement of ischemic cells was *superdiffusive*—that is, significantly more directional than expected by a random walk (Wu et al, 2014) (Fig 3F, **m**ean-**s**quared **d**isplacement (MSD) constant $\sigma$ = 1.19; Fig S3H). In contrast, MSD analysis of cells in well-nurtured clusters was indistinguishable from a random walk (Fig 3F, MSD constant, $\sigma$ = 1.02; Fig S3H).

To further explore these data, we defined *runners* as cells whose net displacement was greater than an arbitrary distance threshold (50 $\mu m$). At most values of this threshold, ischemic clusters had about twice as many runners than well-nurtured spheroids or spheroids in chambers without consumer cells. Interestingly, we observed a directional bias in the movement of runner cells away from ischemia as if they were moving toward nutrient sources (Figs 3G and S3I). However, this directional bias was small and only

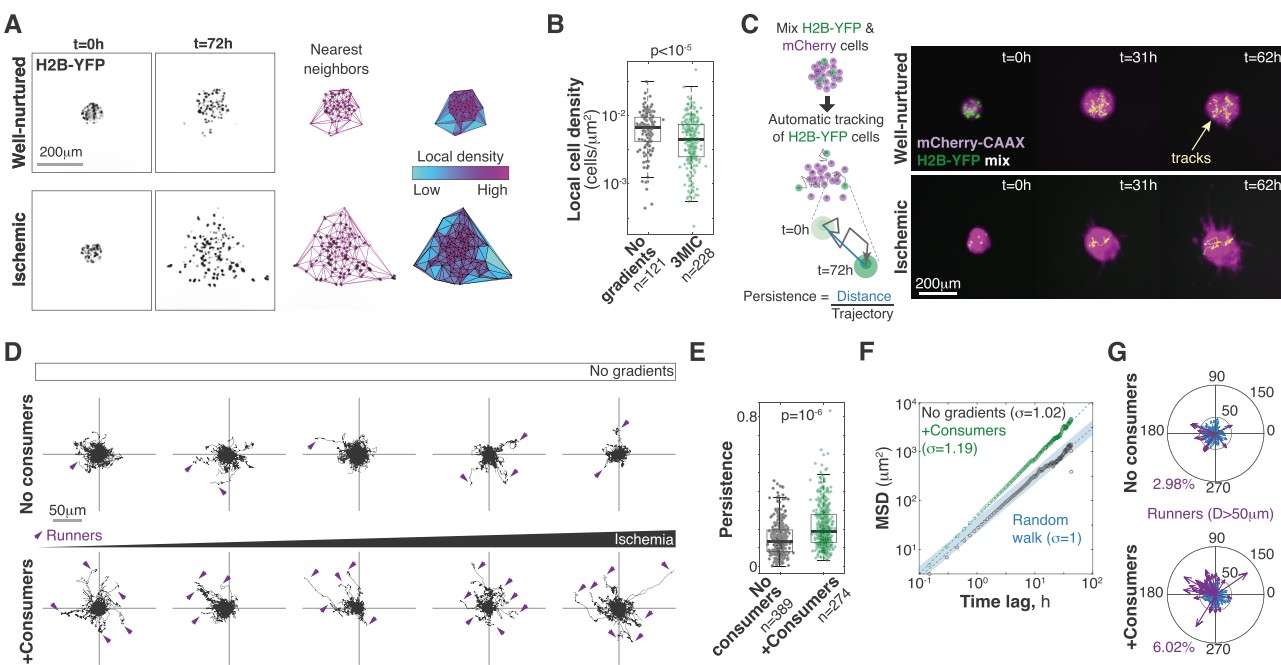

**Figure 3. Ischemic cells in the 3MIC migrate more persistently.**
**(A)** Image-based method to estimate local cell density. We located individual cells expressing the fluorescent nuclear marker H2B-YFP. We then triangulated the closest neighbors and estimated the areas between them. Local cell density corresponds to the inverse of each triangle's area. **(B)** Quantification of local cell densities over time for spheroids in the presence or absence of consumer cells. Data are from 10 different spheroids (5 per condition). **(C)** Experimental setup and representative data of automatically tracked individual cells. **(D)** Top: migratory tracks of cells from five different tumor spheroids located at increasing distances from the opening of the 3MIC in the absence of consumer cells. Bottom: migratory tracks of cells from five different clusters located at similar distances from the opening as plots above but in the presence of consumer cells. Arrowheads show tracks for runner cells (defined as having a net displacement of 50 $\mu m$ or more). **(E)** Persistence of individual cells. Cells in ischemic spheroids are significantly more persistent than cells in well-nurtured spheroids. Data obtained from single-cell tracks. **(F)** Mean-squared displacement analysis of all tracked cells. Ischemic cells disperse more than what would be expected from a random walk (mean-squared displacement constant $\alpha > 1$). **(G)** Compass plots showing the direction and magnitude of final displacement for individual cells. Runner cells are highlighted in purple. 3MIC opening (higher nutrients/oxygen levels) is to the left. Data are from a representative experiment of three biological replicates.

significant under some definitions (distance thresholds) of runner cells (Fig S3J). We thus concluded that—at least in this setting—metabolite gradients do not act as significant orientation cues. Still, our data show that ischemia leads to more persistent movements in three-dimensional tumor cultures resulting in highly dispersive and invasive tumor spheroids.

## Ischemic tumor cells lose epithelial features and increase ECM degradation

We then sought to determine specific cell changes triggered by ischemia. From a biophysical standpoint, the MSD changes observed in our cell tracking analyses suggest that the movements of ischemic cells are less constrained (Mardia & Jupp, 2000). Typically, constraints in cell migration are imposed by the ECM or by adhesion to other cells (Fraley et al, 2015; Collins et al, 2020). Consistently, hypoxic tumor cells are known to increase ECM invasion (Rankin & Giaccia, 2016; Moon et al, 2021) and to have lower levels of epithelial cell adhesion molecules such as E-cadherin (E-Cad) (Scheel et al, 2007; Polyak & Weinberg, 2009; Chen et al, 2010; Jiang et al, 2015; Lehmann et al, 2017; Dongre & Weinberg, 2019; Padmanaban et al, 2019). We thus wanted to test whether ischemic spheroids increase ECM degradation and/or whether they show changes in cell–cell adhesion. To measure ECM degradation, we first used an assay

where ECM proteins are laid over a coverslip coated with fluorescent gelatin (Ros et al, 2020). More invasive clusters will break down the ECM network, and eventually degrade the labeled gelatin resulting in local loss of fluorescence (Fig 4A). With this assay, we observed that ischemia triggered a large increase in ECM degradation in tumor spheroids (Fig 4B) and in consumer cells (Fig S4A). As an orthogonal approach, we cultured tumor spheroids in DQ collagen—a fluorescent but quenched form of collagen that lights up when cleaved (Sloane et al, 2006). In this assay, ischemic spheroids also showed significantly higher levels of ECM degradation and foci of ECM degradation often co-localized with protrusive regions within spheroids (Fig S4B).

Metabolic stress in the tumor microenvironment can decrease epithelial features such as E-Cad levels in carcinomas as we have previously shown in the MEMIC (Janská et al, 2021). Consistent with these previous data, immunofluorescent analysis of ischemic spheroids in the 3MIC showed a dramatic decrease in E-Cad levels (Fig 4C). We observed a similar trend using a fluorescent protein driven by the promoter of E-Cad (Fig S4C), and thus suggesting that ischemia regulates E-Cad at the transcriptional level. Lung KP cells show an intermediate epithelial-to-mesenchymal transition (EMT) status, as denoted by the co-expression of epithelial markers including E-cadherin and mesenchymal markers such as vimentin (Fig S4D). Although in our experiments we see a decrease in

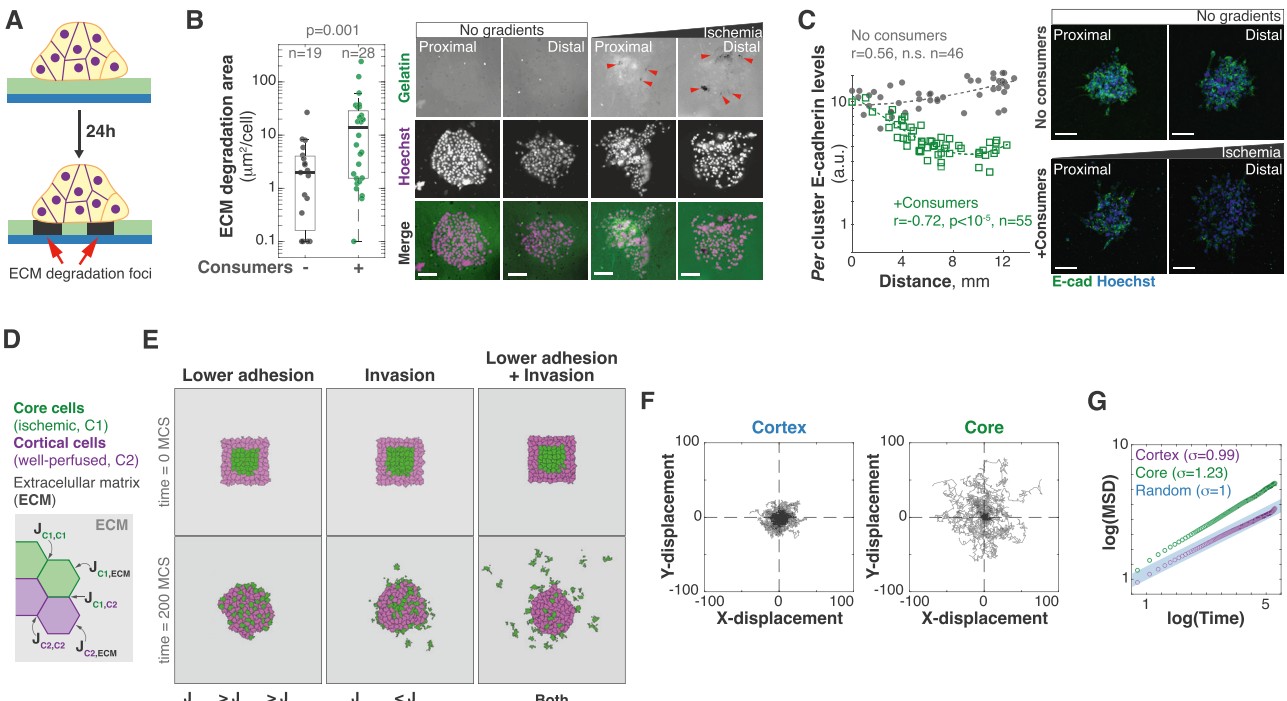

**Figure 4. Decrease in epithelial adhesion and increased invasion can synergize to efficiently disperse tumor cells.**
**(A)** Schematic depicting the ECM degradation assay where spheroids are grown on a glass coated with fluorescent gelatin. Local loss of fluorescence indicates ECM degradation foci. **(B)** Representative images and quantification of ECM degradation. Spheroids were grown in the presence or absence of consumer cells. Ischemic spheroids show higher gelatin degradation (dark spots, red arrowheads). Bars: 100 $\mu$m. Images compare spheroids located 2 mm or closer to the opening of the 3MIC (Proximal) or further than 8 mm (Distal). Data are from a representative experiment of three biological replicates. **(C)** Representative images and quantification of Lung KP spheroids grown in the 3MIC and stained for E-cadherin (E-Cad, green). Nuclei are stained with Hoechst (blue). Bars: 100 $\mu$m. Images compare spheroids located 2 mm or closer to the opening of the 3MIC (Proximal) or further than 8 mm (Distal). Data points: *per* spheroid E-Cad levels. Dashed line: linear fit. Data are from a representative experiment of three biological replicates. **(D)** Schematic depicting cell–cell and cell–substrate interaction energies from our cellular Potts model. See more details about the model in the main text and in the Supplementary Information. **(E)** Initial and final conditions of simulated tumor populations moving under different rules. The combination of lower epithelial adhesion and invasion, but not each process alone, efficiently increases the dispersion of ischemic (core) cells. **(F)** Tracks of simulated ischemic (core) and well-nurtured (cortical) cells modeled to have lower epithelial adhesion and increased invasion. **(G)** MSD of simulated well-nurtured and ischemic cells with lower epithelial adhesion and invasion. MSD changes are in the same order of magnitude as in our experimental data (Fig 3F).

epithelial features, we did not observe significant changes in the levels of vimentin along the 3MIC, suggesting that ischemia promotes a partial EMT (Fig S4E). Interestingly, a recent study demonstrated that the partial EMT in vimentin-positive carcinoma is required for metastasis in triple-negative breast cancer preclinical models (Grasset et al, 2022). In summary, these results show that ischemic cells decrease epithelial adhesion and increase the degradation of the surrounding ECM.

## A model of persistent tumor migration in the absence of directional cues

Our results are consistent with well-established evidence that tumor ischemia increases migratory and invasive cell properties (Quail & Joyce, 2013; Rankin & Giaccia, 2016; Nobre et al, 2018; Godet et al, 2019). Because these conditions arise deep within tumors, how do nascent metastatic cells move through the tumor tissue to eventually reach the tumor's edge? An appealing hypothesis is that nutrients or oxygen acts as directional cues that orient the movements of ischemic cells. However, we did not find enough evidence in our data to support this hypothesis (Figs 3G and S3H

and I). We thus asked whether our other observations: a decrease in epithelial adhesion and increased ECM degradation, are sufficient to allow the escape of ischemic cells from the nutrient-deprived regions they emerge from.

We decided to address this question conceptually using a cellular Potts model (CPM) (Graner & Glazier, 1992; Glazier & Graner, 1993)—a mathematical framework frequently used to model problems of differential cell adhesion and cell sorting in morphological processes (Szabó & Merks, 2013). In a CPM, any number of "cell types" exist on a grid and have different affinities for each other and for their surrounding substrate. High-affinity contacts have a lower surface energy, and thus, they are more stable. In contrast, low cell–cell or cell–substrate affinities are represented as highly energetic and unstable contacts. At every simulated time point (a Monte Carlo step), the model tries to minimize the energy of the system resulting in changes in cell shape, localization, etc., according to local energy levels (Graner & Glazier, 1992; Glazier & Graner, 1993).

Our modeled tumors were formed by two cell types: *core* (ischemic) and *cortical* (well-perfused) cells designated according to their positions at the beginning of the simulation. Our specific goal

was to test whether changes in cell–cell and cell–substrate adhesion would allow core cells to move through cortical cells and eventually invade the surrounding ECM. We modeled the reduced levels of epithelial adhesion we observed in ischemic cells as a decreased cell–cell affinity by increasing the surface energy between core cells, that is, decreasing their stability. Energy levels were set to ensure that the adhesion between core cells was low, between cortical cells was high, and between the two cell types was intermediate. We tested a range of energy parameters obtaining similar qualitative results as long as we kept these relationships (Fig 4D; see additional details in Tables S1 and S2). Implementation of these changes resulted in clusters where core cells mixed with cortical cells, but there was no invasion into their surroundings (Fig 4E; Video 5). To model the increased ECM invasion in ischemic cells, we increased the affinity for the surrounding ECM in core cells. The intuition behind this step is an increase in ECM degradation will facilitate the movement of tumor cells into the ECM, which in a CPM equates to a higher ECM–tumor affinity (Fig S4F). This feature alone produced poor mixing between populations and a moderate trend to invade the ECM (Fig 4E; Video 5). However, cell movements were random and did not show the persistence we observed in our experimental data (Fig S4G).

The combination of these two rules, however—core cells with decreased cell–cell adhesion and increased affinity for the ECM—produced the most interesting outcomes. Ischemic core cells moved through layers of well-perfused cells and rapidly dispersed through the surrounding ECM (Fig 4E; Video 5). These virtual cell movements were remarkably close to the movements of our experimental cells. For example, tracks of core and cortical cells in the model (Fig 4F) resembled the movements of ischemic and well-nurtured cells, respectively (compare with Fig S3F). Most strikingly, the MSD of core cells in this scenario increased by the same order of magnitude as in our experimental cells in the 3MIC (compare Figs 3F and 4G). Altogether, these data suggest that increasing the affinity for the ECM and decreasing cell–cell adhesion are sufficient for the spontaneous emergence of directional movements driving the dispersion of core cells into the ECM—even in the absence of explicit directional cues. Although many other mechanisms will contribute to the translocation of ischemic tumor cells to regions from where they can spread in vivo, this model proposes a minimal set of conditions that can facilitate this process.

## Pro-metastatic features induced by ischemia are reversible

Hypoxic and ischemic environments can select for more aggressive and metastatic clones (Labuschagne et al, 2019; Godet et al, 2019; Tasdogan et al, 2021). The contribution of clonal selection in our brief 3MIC experiments (~24–72 h) is negligible. In vivo however, timescales are much longer, which may allow ischemia to select for more migratory clones. We hypothesized that if ischemic cells are selected to be more migratory in vivo, they should display a higher migratory phenotype in the 3MICs—irrespective of their location along the ischemic gradient. We thus injected Lung KP cells expressing HRE-GFP into the flank of syngeneic mice. After 2 wk, we extracted these tumors and dissociated them into single cells. Using flow cytometry, we sorted ischemic from non-ischemic cells according to their GFP levels (Fig S5A). From these sorted cells, we

generated tumor spheroids, which we then cultured into separated 3MICs (Fig 5A; Fig S5B). As shown in Fig 5, the increase in cell invasion strongly correlated with the distance to the opening but not with their precedence (Fig 5B and C). Consistently, spheroids formed neither from hypoxic nor from non-hypoxic tumor cells showed an increased invasion when cultured in 3MICs without consumers (Fig S5C and D). These results show that at least in this system, the pro-metastatic features produced by ischemia are phenotypic changes that do not require clonal selection.

## Reconstruction and visualization of more complex microenvironments

Solid tumors are infiltrated by a variety of stromal and immune cells (Condeelis & Pollard, 2006; Quail & Joyce, 2013; Massagué & Ganesh, 2021). Tumor-associated macrophages are one of the most abundant cell types in solid tumors, and their presence correlates with increased cancer progression, metastasis, and mortality (Condeelis & Pollard, 2006; Gocheva et al, 2010; Colegio et al, 2014; Noy & Pollard, 2014; Bronte & Murray, 2015; Wenes et al, 2016; Linde et al, 2018). Tumor-associated macrophages recruit endothelial cells helping to orchestrate the vascularization process required for tumor growth and metastasis (Wenes et al, 2016). We recently showed that a combination of hypoxia and high lactic acid activates the ERK/MAPK pathway in macrophages which then secrete VEGFA and induce tube-like morphogenesis in endothelial cells (Carmona-Fontaine et al, 2017). To test whether we could recapitulate these data in the 3MIC, we generated endothelial clusters from SVEC4-10 cells (referred to as SVECs) and co-cultured them with bone marrow-derived macrophages (BMDMs). As controls, we examined similar SVEC clusters in 3MICs without macrophages. In the absence of macrophages, clusters of endothelial cells remained mostly rounded, even in the presence of consumer cells and regardless of their location along the ischemic gradient (Fig 6A). In the presence of macrophages, however, ischemic endothelial cells extended away from clusters and sprouted into the ECM (Fig 6A; Video 6). This tube-like morphogenesis did not occur in well-nurtured regions of the 3MIC (Video 6). We did not notice that the cell type used as consumers produced significant differences in endothelial sprouting, but it always required the infiltration of macrophages (Fig S6A). Consistent with previous evidence (Wenes et al, 2016; Carmona-Fontaine et al, 2017), inhibition of VEGF signaling with linifanib abrogated the endothelial sprouting induced by macrophages (Figs 6B and S6B). Although this inhibitor may have off-target effects, these data are consistent with prior data showing VEGF secretion and a pro-angiogenic role in ischemic macrophages (Wenes et al, 2016; Carmona-Fontaine et al, 2017).

We then co-cultured macrophages with tumor spheroids. In these experiments, we observed that the presence of macrophages increased the migration and ECM invasion of tumor cells. This increase was evident even in well-nurtured regions, but much stronger under ischemia (Fig 6C). Similarly, macrophages and ischemia synergized to decrease E-Cad levels (Fig 6D).

Live imaging of these tumor–macrophage interactions also revealed a fascinating behavior: tumor cells from ischemic spheroids extended protrusions toward macrophages appearing to physically drag them into the tumor cell cluster (Fig 6C; Video 7).

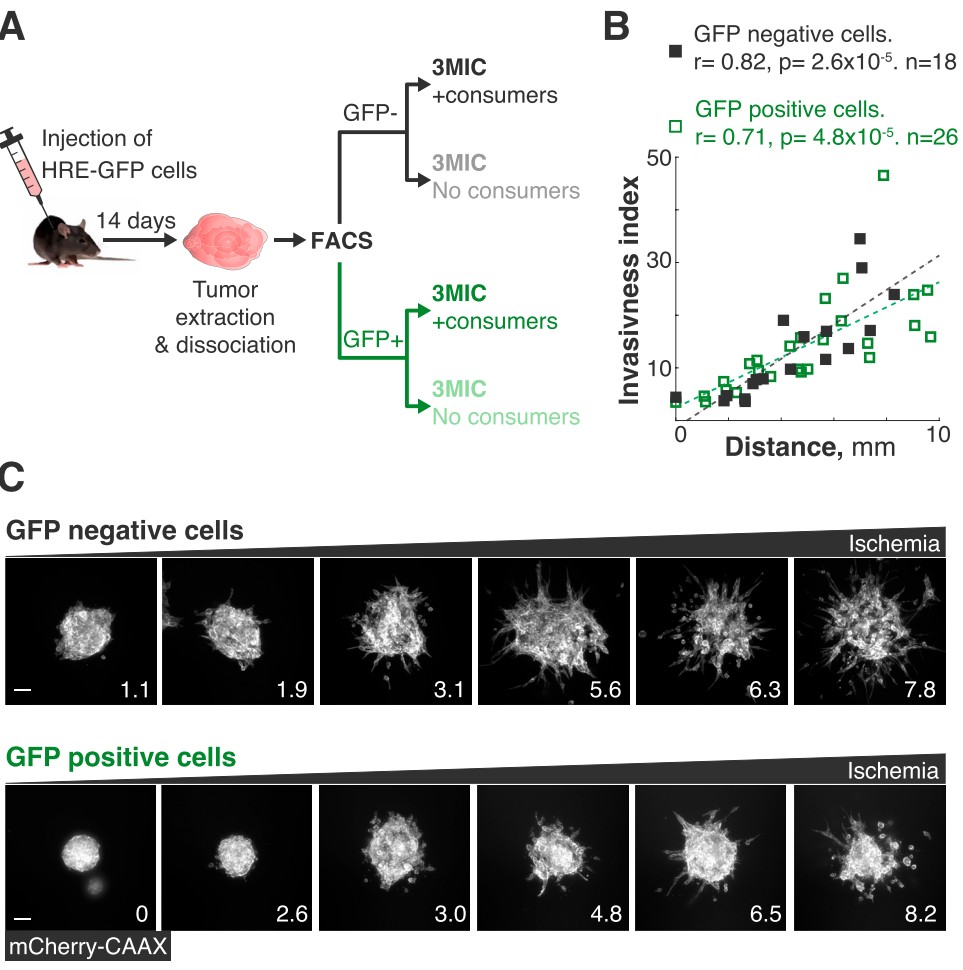

**A** Injection of HRE-GFP cells

14 days

Tumor extraction & dissociation → FACS

GFP-
3MIC +consumers
3MIC No consumers

GFP+
3MIC +consumers
3MIC No consumers

**B**
GFP negative cells.
r= 0.82, p= 2.6x10$^{-5}$. n=18

GFP positive cells.
r= 0.71, p= 4.8x10$^{-5}$. n=26

Invasivness index

Distance, mm

**C**
**GFP negative cells**
Ischemia
1.1  1.9  3.1  5.6  6.3  7.8

**GFP positive cells**
Ischemia
0  2.6  3.0  4.8  6.5  8.2
mCherry-CAAX

**Figure 5. Tumor hypoxia in vivo did not induce permanent changes in the migratory potential of tumor cells.**
**(A)** Lung KP cells expressing HRE-GFP and mCherry-CAAX were injected subcutaneously in C57BL/6 mice and allowed to grow for 14 d. Tumors were then extracted and sorted into GFP-positive (hypoxic/ischemic) and GFP-negative (normoxic/well-nurtured) populations. These different subpopulations were grown as tumor spheroids in 3MICs in the absence or presence of consumer cells. **(A, B)** Quantification of invasiveness of the experimental setup shown in (A). Data points: the invasiveness index of individual spheroids. Dashed lines: linear fit. We observed similar invasion levels and response to ischemia in all spheroids regardless of whether they were derived from well-nurtured or ischemic tumor regions. **(C)** Representative images of tumor spheroids derived from GFP-positive and GFP-negative cells growing at different locations inside of 3MICs. Numbers in the lower right corner denote the distance to the opening in mm. Bars: 100 μm.

This observation suggests that there may be an understudied mechanical interaction where malignant cells may "pull" macrophages into the tumor. Increasing the complexity of co-cultures in the 3MIC is simple. For example, we produced a triple co-culture of chimeric SVEC/Lung KP spheroids grown in the ECM and surrounded by macrophages (Fig 6E; Video 8). Here again, we observed the strong synergy between metabolic and macrophage signals in promoting morphological changes in tumor and endothelial cells.

### Using the 3MIC to test the effects of anti-motility drugs on ischemic cells

Limited diffusion of anti-cancer drugs from blood vessels into solid tumors is a major therapeutic challenge (Tannock et al, 2002; Minchinton & Tannock, 2006) (Fig 7A). Multiple factors account for poor drug penetration including tumor vascularization and the distance between blood vessels (Jain, 1997), simple diffusion through cells and the ECM (Brown et al, 2003), and drug consumption and degradation by cells in the tumor microenvironment (Shree et al, 2011). In addition, hypoxic and ischemic cancer cells can develop drug resistance by increasing ABC transporters and drug efflux (Robey et al, 2018), and bypass drug targets through metabolic rewiring (Pranzini et al, 2021). Distinguishing biophysical limitations to drug penetration from the acquisition of true drug resistance is necessary to improve therapy, and yet, most assays struggle to untangle these factors (Minchinton & Tannock, 2006). The 3MIC can solve some of these issues providing a unique opportunity to test drugs that directly target metastatic initiation.

To illustrate this principle, we treated tumor spheroids with anti-migratory doses of Taxol. At high doses, Taxol stabilizes microtubules leading to mitotic arrest and cell death (Weaver, 2014). However, sublethal concentrations of this drug inhibit migration and invasiveness in several cell types (Belotti et al, 1996; Wang et al, 2019) (Fig S7A). In our experiments, we determined that concentrations of 20–200 nM of Taxol drastically inhibited the movements and invasion of well-nurtured Lung KP spheroids with no effects on their cell viability (Fig S7B). Similar levels of this drug, however, had much milder effects or no effects at all on the movements of ischemic tumor spheroids (Fig S7C).

To distinguish whether these differences are due to a decrease in Taxol levels along the 3MIC, or whether they denote a differential response in ischemic cells, we took advantage of fluorophore-conjugated Taxol analogs. These compounds are routinely used to label microtubules, and thus, we can use the intensity of microtubule staining as a proxy for local drug levels (Figs 7B and S7D). After adding known quantities of the fluorescent Taxol analog

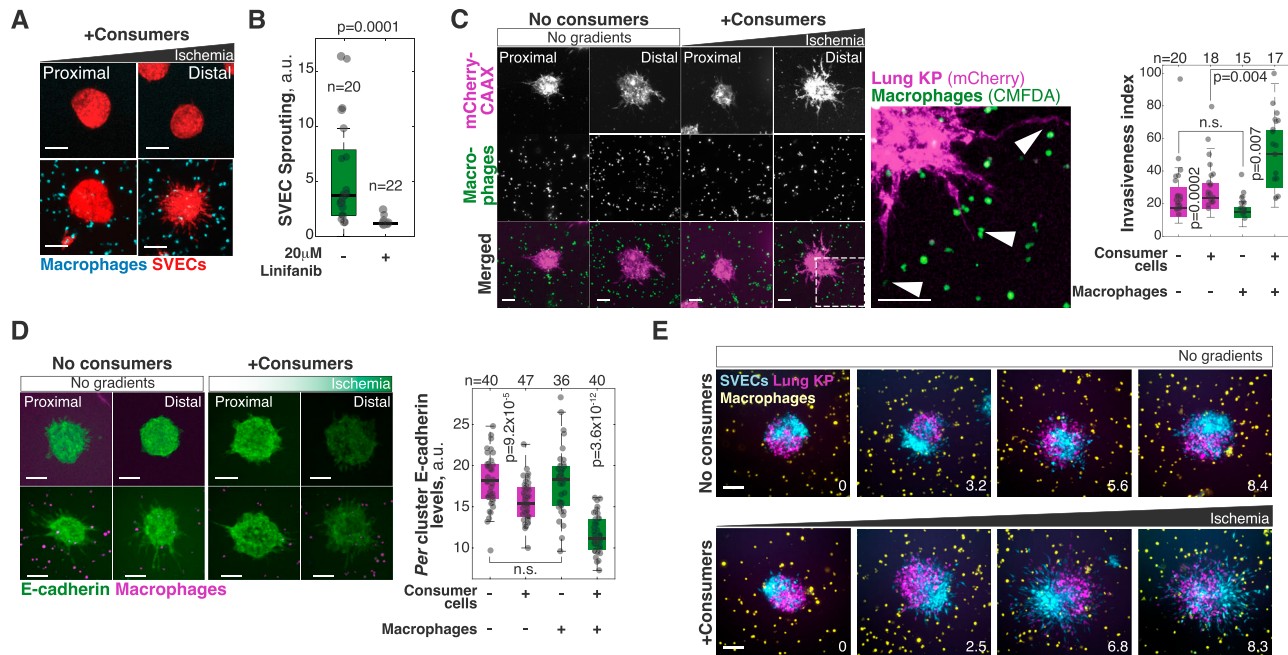

**Figure 6. 3MIC allowed the seamless implementation of cell co-cultures.**
**(A)** Clusters of endothelial cells (SVECs, red) grown in 3MICs in the presence or absence of BMDMs and stained with CMFDA (cyan). Bars: 100 μm. Images compare spheroids located 2 mm or closer to the opening of the 3MIC (Proximal) or further than 8 mm (Distal). Data are from a representative experiment of four biological replicates. **(B)** Quantification of SVEC sprouting in clusters co-cultured with macrophages. VEGFA inhibitor linifanib abrogates sprouting induced by ischemic macrophages. Data are from a representative experiment of two biological replicates. **(C)** Representative images and quantification of invasiveness of Lung KP spheroids co-cultured with BMDMs in the 3MIC. Ischemic macrophages significantly enhance tumor invasiveness and synergize with ischemia. The region delineated with dashed lines is magnified on the right. Data points: *per* spheroid invasiveness. Bars: 100 μm. Images compare spheroids located 2 mm or closer to the opening of the 3MIC (Proximal) or further than 8 mm (Distal). Data are from a representative experiment of three biological replicates. **(D)** Representative images and quantification of Lung KP spheroids grown in the 3MIC and stained for E-cadherin (green) in the presence or absence of macrophages (magenta). Ischemia and the presence of macrophages synergize in their reduction of E-cadherin levels. Bars: 100 μm. Images compare spheroids located 2 mm or closer to the opening of the 3MIC (Proximal) or further than 8 mm (Distal). Data are from a representative experiment of four biological replicates. **(E)** Representative images of triple co-cultures: chimeric Lung KP cells and SVEC spheroids were grown in the presence of macrophages with or without consumer cells. The numbers in the lower right corner denote the distance to the opening in mm. Bars: 100 μm.

(Taxol-Cy5), fluorescence levels progressively decreased in distal spheroids, but they did so at the same rate in 3MICs with or without consumer cells (Figs 7C and S7D and E). These results indicate that consumer cells have a negligible effect on the Taxol gradient. From the changes in microtube staining intensity, we determined that in a 3MIC culture treated with 150 nM of Taxol, the most ischemic spheroids will experience about ~30 nM of Taxol (45 nM at 10 mm and 24 nM at 12 mm; Fig 7C). Although these Taxol levels have no effects on ischemic tumor spheroids, 30 nM of this drug completely inhibited the movements of well-nurtured tumor cells (Fig 7D). We thus concluded that ischemic spheroids are more resistant to Taxol by mechanisms beyond drug penetration and diffusion.

## Discussion

Animal models are fundamental to study the complexity and heterogeneity of the tumor microenvironment (Day et al, 2015; Gould et al, 2015). In these in vivo models, however, it is hard to isolate and weigh the contributions of different variables to tumor progression, and large experiments are prohibitively expensive for most laboratories. On the contrary, conventional in vitro approaches offer

much better experimental control and can be easily scaled up for high-throughput approaches. However, in vitro models do not usually consider the cellular and molecular heterogeneity of the tumor microenvironment. The lack of models that can bridge the complexity of in vivo models with the ease of in vitro cultures has impaired progress in understanding the tumor microenvironment and how it can modulate the emergence of metastatic tumor features (Fontebasso & Dubinett, 2015; Anderson et al, 2019; Ganesh & Massagué, 2021). To bridge this technical gap, we described the design and implementation of the 3MIC—an ex vivo culture system that allows visualizing the acquisition of metastatic-like migratory properties in complex 3D tumor cultures. Altogether, the experiments presented here show that the 3MIC recapitulates key features of the tumor microenvironment. It allows for imaging ischemic cells that are hidden from conventional techniques and permits testing the role of different elements in the tumor microenvironment and the effects of drugs on the emergence of metastatic features (Fig 7E).

The results shown here are consistent with well-established evidence that hypoxia is essential for the emergence of metastasis. The 3MIC, however, allowed us to gain new insights into this process by (1) recreating a complex ischemic-like metabolic microenvironment rather than simple hypoxia and (2) overcoming challenges associated with imaging ischemic cells that are usually located deep within

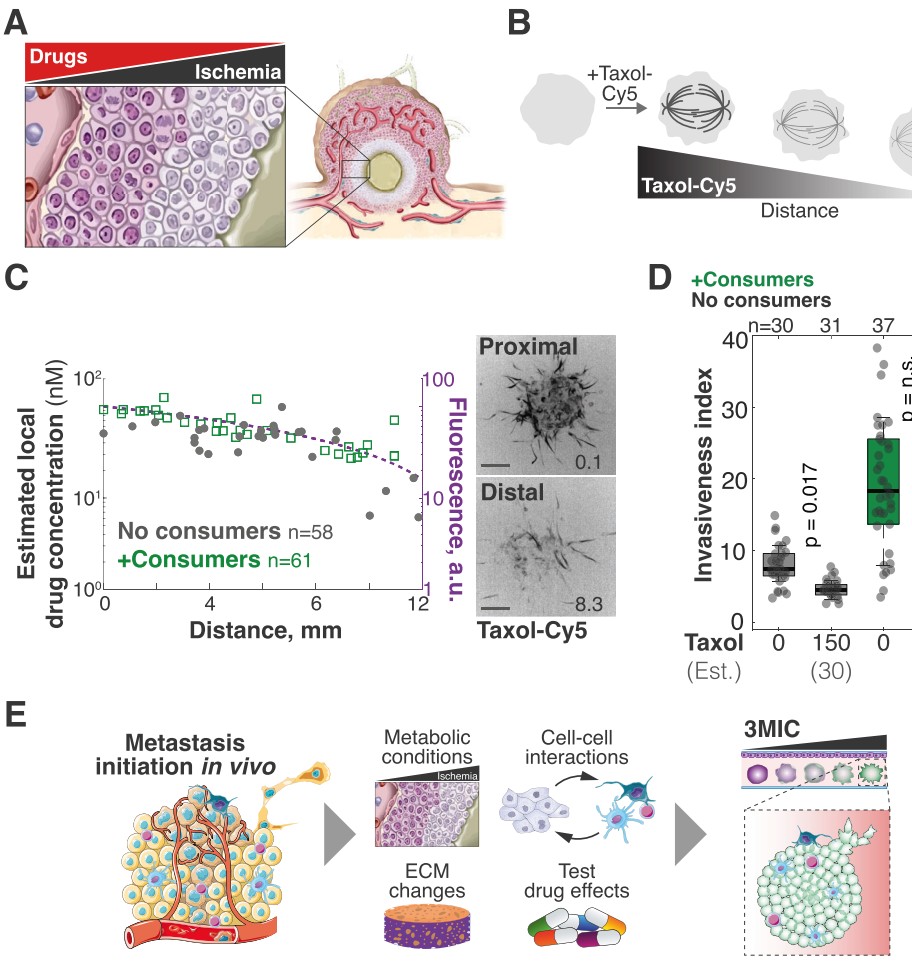

**Figure 7. 3MIC allows studying the effects of anti-migratory drugs on ischemic cells.**
**(A)** Cartoon showing that drug penetration of tumors is a therapeutic challenge, especially in ischemic regions where drug levels are likely lower. **(B)** Taxol-Cy5 is a probe used to fluorescently label microtubules. We used this property to estimate how Taxol levels decrease because of diffusion along the 3MIC. **(C)** Quantification of Taxol-Cy5 in spheroids within 3MICs with or without consumer cells. Plots indicate that drug levels decrease as they diffuse through 3MICs regardless of the presence of consumer cells or not. Dots: Taxol-Cy5 and location of single spheroids. Data are pooled from two biological replicates. **(D)** Invasion of Taxol-treated spheroids growing in 3MICs with or without consumer cells. Cultures were treated with 150 nM of Taxol which we estimated to drop to about 30 nM deeper within 3MICs. Points: invasiveness of individual distal clusters (located at 4 mm or more from the opening). Data are from a representative experiment of three biological replicates. **(E)** Cartoon showing features of the 3MIC as an ex vivo model to study metastasis.

tumor structures. The ischemic-like environment formed in the 3MIC better mimics metabolic conditions in tumors such as accumulation of lactic acid and nutrient deprivation (Vaupel et al, 1989; Hobson-Gutierrez & Carmona-Fontaine, 2018; Janská et al, 2021). In our experiments, we found evidence that medium acidification has a more direct effect on the invasive properties of ischemic spheroids than hypoxia. We think that medium acidification may be the direct signal that triggers invasion under ischemia, whereas hypoxia indirectly contributes to this effect through the pro-glycolytic role of HIF1A (Lum et al, 2007). At this moment, we can only speculate why pH perturbations may have such strong effects in our experiments. A possible explanation is that many ECM-digesting metalloproteases have lysosomal origins or are more active in more acidic environments, and thus, extracellular acidification would increase the efficiency of these enzymes (Rozhin et al, 1994; Robey et al, 2009; Cappellesso et al, 2022). Although our data show a critical role of hypoxia and extracellular pH, additional metabolic (Lehúede et al, 2016; Bergers & Fendt, 2021) and non-metabolic cues are likely to increase cell invasion under ischemia. For example, glucose deprivation triggers the release of pro-migratory cytokines (Püschel et al, 2020) and there is a growing list of metabolites that directly or indirectly affect the number and destination of metastases (Lehúede et al, 2016; Bergers & Fendt, 2021).

Our data show that ischemic cells display a more efficient dispersion. Consistent with previous evidence (Lee et al, 2005; Chaturvedi & Kaczmarek, 2014), we showed here that ischemia decreases epithelial cell adhesion and increases the degradation of the ECM. Although some of our observations share features with the process of the EMT, we preferred referring to a partial EMT as we only observed a decrease in epithelial features without a significant change in mesenchymal markers. In addition, the contribution of EMT to the metastatic process remains controversial (Chen et al, 2010; Fischer et al, 2015; Padmanaban et al, 2019). An additional issue to consider is the composition of the ECM as they can alter the phenotypes and behavior of tumor and stromal cells (Zhang et al, 2011). Although in our experiments we used a simple combination of collagen and Matrigel, the 3MIC allows testing different ECM compositions and structural scaffolds (Fig 7E).

At the single-cell level, ischemia increases cell persistence without a significant change in cell polarity or increasing directional cell migration. However, we observed a small bias toward higher nutrients in the direction of cell movements. Although in our setting these differences were not statistically significant, this observation would be consistent with a widespread set of examples of nutrient-driven chemotaxis in prokaryotes and in other eukaryotic cells (Wadhams & Armitage, 2004; Gerisch, 1982). For example, the key

nutrient sensor mTORC2 is critical in neutrophil chemotaxis. Still, additional experiments will be needed to establish whether or not tumor cells can follow nutrient gradients.

Recent evidence shows that tumor hypoxia can act as an evolutionary pressure that selects for tumor clones that are more resilient to stressors such as reactive oxygen species (Godet et al, 2019; Labuschagne et al, 2019; Padmanaban et al, 2019). However, our data show that hypoxia and ischemia can alter cells directly without the need for clonal selection. These two ideas are not in conflict, and our experimental conditions are not free of caveats. Most likely, the acquisition of metastatic features in patients occurs by a combination of selection and opportunistic adaptations to the tumor microenvironment.

The direct visualization of tumor–stroma interactions afforded by the 3MIC has unique advantages. For example, we were excited to observe that the movement of macrophages toward tumor structures seemed to be driven by tumor cells mechanically dragging macrophages into their clusters (see, e.g., Video 7). It is possible, then, that immune infiltration of tumors is aided by physical recruitment by cancer cells. Although we are not aware of previous reports of this behavior, this may be due to the challenges of observing tumor–macrophage interactions in vivo, especially in ischemic tumor regions.

It has been difficult to discern whether a decrease in drug response in vivo is due to lower drug concentration or due to intrinsic changes that make cells more drug-resistant (Dewhirst & Secomb, 2017). This is particularly true in more fibrotic and ischemic tumors such as pancreatic ductal adenocarcinomas (Teicher, 1994; Rohwer & Cramer, 2011). The 3MIC, however, allows for the separation of these effects, and we were able to show that ischemic cells display a true Taxol resistance mechanism. These experiments illustrate the ease of testing drugs on ischemic cells and untangling biophysical factors such as drug diffusion from biological adaptations to anti-cancer drugs. Although the particular example we used benefited from the availability of labeled Taxol analogs, a similar approach is possible using imaging mass spectrometry and other methods that allow the quantification of local drug levels (Alexandrov, 2020).

Finally, the fabrication of the 3MIC is easy and affordable with a conventional 3D printer. This system can seamlessly integrate existing co-culture and ex vivo protocols, and thus, we invite the research community to use the 3MIC when studying metastasis and other processes where the limitation of nutrients in 3D multicellular structures is relevant.

# Materials and Methods

### Cell culture

Lung KP clones and Lung KPKeap1 clones were derived from the $KRAS^{G12D}/TP53^{-/-}$ and $KRAS^{G12D}/TP53^{-/-}/Keap1^{-/-}$ lung adenocarcinoma model, respectively, developed and kindly shared by Dr. Thales Papagiannakopoulos. MCF10A, MCF7, DlD1, and SVEC-4-10 cells were purchased from the ATCC. C6-HRE-GFP cells were a kind gift from Dr. Inna Serganova (Memorial Sloan Kettering Cancer Center, New York, NY, USA). Cells were cultured in high glucose DMEM (#11965-092; Gibco) supplemented with 10% FBS (#F0926; Sigma-Aldrich) at 5% $CO_2$ and 37°C. MCF10A cells were cultured in high glucose DMEM (#11965-092; Gibco) supplemented with 5% horse serum, 20 ng/ml animal-free recombinant human epidermal growth factor (#AF-100-15; PeproTech), 0.5 mg/ml hydrocortisone (#H0888; Sigma-Aldrich), 100 ng/ml cholera toxin (C8052; Sigma-Aldrich), and 10 μg/ml insulin, at 5% $CO_2$ and 37°C. To induce HIF1A signaling in normal oxygen conditions, cell spheroids were treated with 2 mM DMOG (#S7483; Selleckchem) or with 300 μM cobalt chloride (#012303-30; Thermo Fisher Scientific) for 18 h. To alter the pH of the media, we added 25 mM of $NaHCO_3$ (#25080094; Thermo Fisher Scientific) or 20 mM of lactic acid (#L1750; Sigma-Aldrich) to the media.

Lung KP cells stably expressing a hypoxia reporter (pLenti-5XHRE-GFP, #128958; Addgene, denoted here as HRE-GFP), a cell membrane marker (pLenti-mCherry-CAAX, #129285; Addgene), an E-cadherin reporter (pHAGE-E-cadherin-RFP, #79603; Addgene), a cell cycle reporter (pBOB-EF1-FastFUCCI-Puro, #86849; Addgene), LV-YFP (#26000; Addgene), and pLenti-H2B-iRFP720 (#128961; Addgene) were generated using standard lentivirus-mediated stable cell line preparation protocols. H2B-mCherry was cloned by removing YFP from the LV-YFP vector using BamHI and KpnI restriction enzymes. The linearized vector containing H2B was then ligated with mCherry using SLIC (Jeong et al, 2012).

### Formation of 3D tumor spheroids

Cell spheroids were formed via hanging drops following the standard protocols (Vinci et al, 2013). Briefly, cells were dissociated into a single-cell suspension of $10^4$ cells/ml. The cell suspension was distributed into 20 μl droplets onto the lid of a petri dish. The base of the dish was filled with PBS (#14040-133; Gibco) or distilled water to prevent droplet evaporation during the incubation time. The lid was then inverted onto the dish and incubated for 96 h, to ensure the formation of compact spheroids. Typically, we cultured ~50 spheroids in each 3MIC.

### Isolation and differentiation of BMDMs

BMDMs were extracted from C57BL/6 mice following the standard protocols (Carmona-Fontaine et al, 2013, 2017). After isolation of the bone marrow, cells were cultured in low attachment culture dishes (#25384-342; VWR) in high glucose DMEM supplemented with 10% FBS and 10 ng/ml recombinant mouse CSF-1 (#416-ML; R&D Systems) for 7 d.

### Establishment of tumor-derived clones

Approximately 5 x $10^4$ Lung KP cells stably expressing pLenti-5XHRE-GFP and pLenti-mCherry-CAAX were injected subcutaneously into C57BL/6 mice and grown for 14 d. Tumors were then extracted and dissociated using a solution containing 2 U/ml dispase and 4 mg/ml collagenase IV. Tumor cells were enriched as a CD45-negative population and sorted according to their GFP levels. GFP-positive and GFP-negative cells were grown separately as clusters using the hanging drop method described above.

## Generation of Hif1a KO cells

Hif1a knockout cells were generated using CRISPR/Cas9 genome editing. Briefly, forward and reverse sequences (CACCGA-GATGTGAGCTCACATTGTG and AAACCACAATGTGAGCTCACATCTC) were synthesized to form single guide RNAs targeting the Hif1a gene. A vector (lentiCRISPR-v2-puromycin, #98290; Addgene) carrying this single guide RNA was introduced into HEK293T cells together with envelope and packaging plasmids (VSV-G and Delta-8.9) via transfection using Lipofectamine (#L3000008; Thermo Fisher Scientific). Target cells were then infected with collected lentiviral particles, selected using puromycin (10 µg/ml, #A1113803; Gibco), and separated into single clones through limiting dilution.

## 3D printing and microfabrication

To create the framework of the 3MIC, fused filament fabrication was used. Designing of the framework was done using OpenSCAD (https://openscad.org). 3MICs were printed in two parts that were later assembled: the main framework and the upper coverslip holder (where consumer cells are grown). These parts were printed in an Ultimaker 3B printer using a black PLA filament. Alternatively, the framework can be printed using stereolithography printers. In this case, the entire framework (the upper coverslip holder and the main framework) can be printed as one unit. We used Dental SG resin to print the framework on Form 3 (Formlabs). The uncured resin was removed from the prints by washing them in isopropanol for 1 h and further post-curing it with UV irradiation heated at 60°C. A glass coverslip (No. 1, 22 × 60 mm, #48393-070; VWR) was glued to the base of the main framework using a UV-curable adhesive (NOA68; Norland Products, Inc.) using a long-wave UV lamp (DR-5401; MelodySusie). Then, the coverslip holder fragment was glued onto the glass coverslip. Finally, the assembled 3MIC was sterilized in a short-wave UV lamp chamber (CM-2009; Meishida) for 15 min.

## Culture of tumor spheroids in the 3MIC

To prepare the layer of consumer cells, 24 h before the day of the experiment, a coverslip (#48366-045; VWR) was coated with poly-D-lysine (#P6407; Sigma-Aldrich) to aid adherence of cells to the glass surface. Cells were detached using trypsin and diluted into a cell suspension at a density of $3.5 \times 10^5$ cells/ml for faster dividing cells and $10^6$ cells/ml for slower dividing cells. The coverslip was placed into a well of a six-well dish and covered with 1 ml of the cell suspension. The coverslip was incubated overnight at 37°C to let the cells adhere to the coverslip. On the day of the experiment, the spheroids were collected from the petri dishes by inverting the lid and gently flushing all the spheroids with DMEM supplemented with 10% FBS using a 5-ml serological pipette. The spheroid-containing suspension was collected and centrifuged at 50$g$ for 10–15 min. Meanwhile, a thin bed of 110 µl ECM was made in the 3MIC—here, we used either 2.2 mg/ml rat tail collagen (#354236; Corning) or 2:3 mix of Matrigel and 1.6 mg/ml of rat tail collagen I—and incubated at 37°C for 20 min to allow the polymerization of the ECM. Spheroids were resuspended in about 110 µl of the ECM and immediately transferred onto the polymerized ECM bed. The ECM with spheroids was polymerized at 37°C for ~30 min. With the help of forceps, the

coverslip containing the consumer cells was inverted and inserted into the slot of the coverslip holder. The 3MIC was then gently filled with 1.25 ml of media, avoiding trapping bubbles underneath the consumer coverslip. Control wells are assembled in the same way, but the top coverslip has no consumer cells.

## Collagen degradation assay

To investigate the invasive potential of cells across the gradient, the ECM was mixed with DQ collagen, type I from Bovine Skin, Fluorescein Conjugate (#D12060; Invitrogen) at a final concentration of 25 µg/ml. This ECM was used to prepare the bed and embed the spheroids as well. The extent of degradation of the matrix was determined by measuring the fluorescence intensity of cleaved collagen.

## Gelatin degradation assay

Glass coverslips were coated with gelatin from pig skin, Oregon Green 488 Conjugate (#G13186; Thermo Fisher Scientific), by inverting coverslips on 20 µl drops of fluorescent gelatin (1 mg/ml) and incubating for 20 min. The coverslips were then lifted, the excess liquid was drained, and they were inverted on 40 µl drops of 0.5% glutaraldehyde for fixation. The coverslips were then washed with PBS and glued onto the PLA frame using the UV-curable adhesive (#NOA68; Norland Products), and cured in a long-wave UV lamp (#DR-5401; MelodySusie) for 15 min. The cured coverslip was then UV-sterilized and used for the experiment. Briefly, the spheroids were collected from the hanging drops and centrifuged and resuspended in 100 µl of media. The spheroid solution was carefully placed on the coated coverslips and incubated for 30–60 min to allow the spheroids to settle down. After ensuring that there are no floating spheroids, the consumer coverslip was transferred into the 3MIC and the remaining 1.4 ml of media was added. The 3MICs were incubated at 37°C for 24 h. After 24 h, the 3MICs were fixed using 4% PFA for 10 min and imaged using a 10X objective of a Nikon Eclipse Ti2-E spinning disk confocal microscope.

## Microscopy

Microscopic images were captured using a 10x objective of a Nikon Eclipse Ti2-E spinning disk confocal microscope, at fixed time points, or set up for time-lapse microscopy. For time-lapse microscopy, cells were maintained at 37°C using a stage-top incubator (#INUCG2-KRi; Tokai Hit) and the chamber was connected to a humidified mixed gas source (5% $CO_2$). Spheroids from different regions of the 3MIC were imaged at suitable time intervals in different fluorescent channels. Typically, we imaged ~10 spheroids per 3MIC every 15–30 min for 24 h. The extracellular pH was estimated using the ratio of the green over the blue channels on RGB images of cell culture media containing the pH indicator phenol red. As pH decreases and the media yellow, the signal on the green channel increases, whereas the blue channel remains largely unchanged. To match this ratio to specific pH values, we imaged and measured the pH of the same culture media supplemented with different levels of lactic acid.

### Immunofluorescence

Spheroids within the 3MIC were fixed using 4% PFA (#19943; Affymetrix) for 10 min at room temperature and washed four times with PBS. Cultures were permeabilized using PBS containing 0.5% (vol/vol) Triton X-100 (#T8787; Sigma-Aldrich) for 20 min and washed three times with PBS. Slides were blocked in 10% FBS in immunofluorescence (IF) buffer (0.3 M glycine in PBS) for 60 min at room temperature. Primary antibodies were diluted at 1:200 in the same blocking solution. We used anti-phospho-H3 (#9706; Cell Signaling Technologies), E-cadherin (#610182; BD Biosciences), and CD45 (#AF114; R&D) antibodies. 3MICs were incubated with primary antibodies overnight at 4°C, then washed with IF buffer, and incubated overnight at 4°C with secondary antibodies also diluted in blocking solution at 1:200. Cultures were then washed and counterstained with 0.5 µg/ml Hoechst 33342 in PBS (#H1399; Invitrogen).

### Image analysis

Automated cell tracking was conducted as previously described (Carmona-Fontaine et al, 2013). Persistence is defined as the ratio between the net displacements of a cell over its trajectory. We estimated the local cell density using a method based on the Delaunay triangulation of neighboring cells (Carmona-Fontaine et al, 2013, 2017). Per cell, per spheroid, and population levels of fluorescence were estimated using custom image analysis software as previously described (Janská et al, 2021).

### CPM simulations

Simulations were conducted using a MATLAB CPM implementation (Tsuyoshi Hirashima, 2024). Adhesion between core (C1) and cortical (C2) cells was implemented as:

$$J_{C1,C1} > J_{C1,C2} > J_{C2,C2}$$

where $J_{i,j}$ is the surface energy at every contact point between the two cells (Graner & Glazier, 1992). This relationship modeled the decrease in E-Cad levels in ischemic cells. The differential affinity for the ECM in core cells was simulated as:

$$J_{C1,ECM} < J_{C2,ECM}$$

The CPM was implemented in a 200 × 200 grid with 100 initial cells and run for $1.28 \times 10^8$ Monte Carlo simulation steps. Parameters used in the conditions presented here can be found in Tables S1 and S2. MATLAB routines to run the simulations are available upon request.

### Statistics

We conducted at least three biological replicates of all experiments, unless stated otherwise. Unless noted, we used two-tailed $t$ tests to estimate $P$-values between two conditions and Pearson's linear coefficient when testing correlations. For multiple comparisons, we used one-way ANOVA followed by a Tukey–Kramer test. In all plots, error bars show the SD from the mean. In boxplots, center lines show the median and box edges show $75^{th}$ and $25^{th}$ empirical quartiles of the data.

## Data Availability

The use of the 3MIC requires a Material Transfer Agreement, and it is free for non-commercial and academic uses.

## Supplementary Information

## Acknowledgements

We thank all members of the Carmofon Laboratory for critical discussions about this project and the article. This work was supported by awards to C Carmona-Fontaine from the National Cancer Institute of NIH (DP2 CA250005) and the American Cancer Society (RSG-21-179-01-TBE). C Carmona-Fontaine is Pew Scholar in the Biomedical Sciences, supported by the Pew Charitable Trust (00034121). J Garcia was supported by NIH Training Grant (T32GM132037-01).

### Author Contributions

L Anandi: conceptualization, data curation, formal analysis, validation, investigation, methodology, and writing—review and editing.
J Garcia: data curation, validation, investigation, and writing—review and editing.
M Ros: data curation, validation, investigation, and writing—review and editing.
L Janská: data curation, validation, investigation, and writing—review and editing.
J Liu: validation.
C Carmona-Fontaine: conceptualization, resources, data curation, software, formal analysis, supervision, funding acquisition, validation, investigation, visualization, methodology, project administration, and writing—original draft, review, and editing.

### Conflict of Interest Statement

NYU holds a patent for the 3MIC and related systems where C Carmona-Fontaine is the inventor.

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
