## [Reviewer comments · Life Science Alliance]

Direct visualization of emergent metastatic features within an ex vivo model of the tumor microenvironment

Libi Anandi, Jeremy Garcia, Manon Ros, Libuše Janská, Josephine Liu, Carlos Carmona-Fontaine

DOI: 10.26508/lsa.202403053

Corresponding author(s): Dr. Carlos Carmona-Fontaine (New York University)

Review timeline:

Submission Date:	2024-09-19
Editorial Decision:	2024-09-20
Revision Received:	2024-09-27
Editorial Decision:	2024-09-30
Revision Received:	2024-10-03
Accepted:	2024-10-07

Scientific Editor: Eric Sawey

Transaction Report:

Please note that the manuscript was reviewed at Review Commons and these reports were taken into account in the decision-making process at Life Science Alliance.

Review
COMMONS

Full Revision

Manuscript number: RC- RC-2023-01994

Corresponding author(s): Carlos, Carmona-Fontaine

[Please use this template only if the submitted manuscript should be considered by the affiliate journal as a full revision in response to the points raised by the reviewers.]

*If you wish to submit a preliminary revision with a revision plan, please use our "Revision Plan" template. **It is important to use the appropriate template to clearly inform the editors of your intentions.**]*

1. General Statements

Dear Editor and reviewers,

We would first like to sincerely apologize for all the time that it took us to revise our manuscript. We were confident that we could swiftly respond all your concerns, but we had some major roadblocks. While some of them were scientific, the main reason behind our delay was some unexpected changes in our lab personnel. However, we are happy to report that we have finally finished a new version of the manuscript that we think addresses all your concerns.

We would then like to thank all three reviewers for their overall positive and constructive assessment of our work. Many of the criticism were actually quite insightful and inspiring and drove us to reanalyze our data and to design new experiments that we consider they have significantly improved our work.

Reviewer #1

Evidence, reproducibility and clarity:

In this work, Anandi et al. propose an ex vivo model that can be used to recapitulate the in vivo structure of the tumor microenvironment, which allows the observation of morphological and functional changes in tumor cells in a 3D context. Due to the ability of cancer cells to induce hypoxic condition within the TME, authors propose this model to tackle the study of metastasis initiation in vitro. The proposed system successfully displays an ischemic gradient with cells accessing nutrients at different rates, similarly to what happens in solid tumors in vivo. Moreover, in line with the literature, tumor cell migration and invasiveness were promoted by hypoxic conditions. Authors also show that the system could be used to study cell-cell interaction, as co-cultures of macrophages and cancer cells were successfully cultured in the system and studied in the context of tumor hypoxia.

The study proposed is interesting and timely, as cancer cell invasion remains an important area of tumor biology that needs further exploration. The methodology is well explained and proposed in a linear flow. However, the work could benefit from some improvement and changes, as well as from additional experiments. On an important note, authors do not properly refer to the current literature, as several studies on 3D culture systems/chambers have already been studied and developed to investigate the tumor microenvironment, but they are not cited nor referred to in the manuscript. Authors should refer to such literature and explain how this system is different and adds to it.

Major comments:

1. Authors propose this method to study the TME in 3D. When culturing cells with different ECM (Collagen vs. matrigel+collagen I) authors should take into consideration the effect of these materials on different cell types. It is known how collagen and matrigel can differently influence the polarization and phenotype of stromal cells (particularly in regards of fibroblasts - major components of solid cancers - e.g., PMID 21029367), therefore these points should be addressed at least in the discussion.

We completely agree with the reviewer so we added this point (and reference) to our manuscript's introduction (lines 45-46) and discussion (lines 442-445).

2. In addition to the previous comment, matrigel and collagen are also known to alter cancer cell phenotype (e.g., PMID 21029367) and this point should be taken into account.

We completely agree with the reviewer so we added this point (and reference) to our discussion in the main text (lines 442-445).

3. The need for novel 3D systems to study different aspects of the TME in vitro/ex vivo are certainly needed, however they are not inexistent. Authors should address this in the text, as the current literature already started to propose 3D models (including models involving matrigel/collagen in combination with other materials). 3D chambers (of different materials, and with different aims) are being used and designed and can be found in the literature. These works are not cited in the current study at all. For instance, Anguiano 2017; Cavo et al. 2018; Anguiano et al., 2020; Sodek et al. 2008, etc.

We agree so we have now added those references to the main text (line 56-57).

4. Even though the focus is on hypoxia and the achievement of an ischemic gradient in the chamber to allow resemblance of an in vivo tumor, the authors write in line 123 (and also in other parts of the text) that: "these results show that consumer cells in the 3MIC form ischemic gradients that can influence the local metabolic microenvironment experienced by neighboring tumor spheroids". The addition of the use of the PMDS membrane partly supports the claim, however it would be interesting to check whether this is indeed true, by measuring for example the levels of certain metabolites (e.g., glucose, glutamine, glutamate, lactate, aspartate) reached with the system, or pH levels, etc., in presence or absence of the hypoxic gradient/consumer cells.

This is an insightful question and defining the exact composition of this complex ischemic microenvironment is a major ambition of our lab, so we completely agree with the reviewer's comment. However, as the 3MIC was designed specifically for microscopy, measuring specific metabolites it is unfortunately outside its capabilities.

Having said that, and following the spirit of the reviewer's comments, we used microscopy to measure additional signs of metabolic stress. Specifically, we used fluorescent probes to detect changes in intracellular pH (pHrodo, Molecular Probes) and in Redox status (CellROX, Molecular Probes) and glucose (2-NBDG – a fluorescent D-glucose analog). As we explain below, we found exciting results from our pH measurements which led us to additional functional experiments. We are very excited about these new results, and we thank the reviewer for encouraging these experiments. These new results also provide evidence that other parameters in ischemia – and not just hypoxia – change along the 3MIC and can have an impact on tumor cells.

5. When looking at the references presented in the manuscript, authors quote too many review articles, rather than scientific articles. Given the extremely wide literature on cancer metastasis, more of these works should be quoted in this context. For example: in the introduction - text lines 27-38 - only 4 references are research articles, out of 14 references presented in that paragraph.

The reviewer is correct in pointing this out. Our intention was to use reviews on topics that are well established where citing primary research could be unfair to

other contributions. But again, we agree with the reviewer, so we replaced reviews with primary research articles in multiple locations along the manuscript.

6. As authors showed successfully how macrophages and cancer cells can interact in the chamber, recapitulating cell interactions in an in vivo context, it would be very interesting to see whether different consumer cells would induce similar or different changes to the spheroids and the ischemic gradient (for instance using stromal cells or non-tumor cell lines as consumers, instead of cancer cells only), as we know how tumors are a multitude of cell subsets, each contributing to nutrient production, oxygen consumption, etc.

This is a great point. We thought about that very same point and conducted several experiments to test the combinatorial effects of different consumer cells. In broad terms, we did not observe major differences when using different consumer cells. However, we agree that this system may provide compelling opportunities to test the effect of different cell types on each other. Still, for consistency and ease, we conducted most of our experiments using the same cells in both consumers and in spheroids.

In the resubmitted version, we added an experiment where we looked at the sprouting of SVEC endothelial cells using the same cells or Lung KPs as consumers (Fig. S6A).

Minor comments:

1. Studying the early metastatic development/seeding remains a timely quest, however authors should refer to several new studies in which various mouse models are used to study metastasis from different points of view (e.g., PMID 25822788; PMID 36991128; PMID 25171411; PMID 25633981; PMID 34632412; PMID 35921474; etc). Or line 41, three reviews are quoted (refs 27-29), whilst there are several works that could be quoted on metabolism in solid tumors also in the context of metastasis (e.g., PMID 36522548; PMID: 26719539, PMID 34303764). This comment applies to the rest of the text.

We thank the reviewer for their help in processing this vast literature. We were aware of most of those works but some were new to us so thanks again! We have now added these references.

2. The order of the references is not properly presented. In the introduction, the first reference is n. 4 (text line 22), instead of it being reference 1. Moreover, the subsequent literature ref. is number 12 and not number 2. Please revise the order of the references, and position them within the bibliography from first cited to last cited in the text.

We apologize for this confusion. We have now revised all the references and we hope they are correctly formatted and numbered. The origin of this confusion may have been that we had references in the abstract thus their numbering started there

rather than from the introduction. To avoid further confusions, we removed all references from the abstract.

3. Lines 98-104. It would be helpful to the reader to define here what these consumer cells are. Even though it is explained in the methods that the consumer cells are cancer cells, it is important to make it clear in the text, as it could be misleading at times.

We agree with the reviewer although we did not mean to be misleading. As mentioned above, we chose to use the same cells for both: consumers and spheroids and we have now added a new figure to illustrate this point (Fig S6A). Following the advice, we are also including additional text to make the message clearer (lines 107-109).

4. The English grammar and spelling should be revised in some parts, as well as typos and missing words throughout the text (e.g., Line 38, the word "interraction" is misspelled and should be corrected with "interaction". Line 49, the first sentence seems incomplete. Lines 68-69 should be revised as the sentences do not flow well together, probably due to a missing word. In line 77 it should be "presents". Line 341 should be "cannot be explained").

We apologize for these typos and mistakes. We have tried our best to avoid these type of errors in the new manuscript version.

Referees cross-commenting

I find the comments from the other reviewers to be in line with one another as well as with my general assessment. The major and comments of all reviewers should be addressed. The minor comments should be taken into account as well, as they would render the text and the figures more precise. I suggest that 3-6 months to complete the revision process is an appropriate time frame for the authors.

Finally, I strongly encourage the authors to add in the discussion the points and questions raised by all reviewers, as well as to improve the bibliography in terms of organisation, linearity, and state of the art.

Significance:

General assessment:

The work by Anandi et al. offers an additional tool to tackle the issue of studying the tumor microenvironment, in a 3D culture system.

The authors show a model that can be used to study tumor hypoxia in 3D, offering the possibility to study the TME in a more in vivo-like manner without turning to mice models. The development of new tools to study the TME avoiding the excessive use of animals is definitely a timely quest. In addition, the system has the potential to be applied to tackle different biological questions, as the methodology is well explained and could be suitable to many other fields of cancer biology (e.g., drug resistance or uptake). The work is overall

Full Revision

presented in a clear way and the methodology is explained thoroughly and it has the potential to be a useful tool for the study of cancer hypoxia.

However, authors should address how their method could differently impact other cells when applied to other systems. As one major claim is the potential use of this methodology to study the TME, it should be taken into consideration how stromal cells are strongly affected by the ECM, and how certain settings or features of the system may impact such cell populations. In addition, the work does not properly refer to the current state of the art. As other studies started to propose 3D systems for the study of TME and cell-cell interactions - besides organoids - the authors should cite these works and frame their own study in a more appropriate context, pointing out differences with the current 3D chambers available, the advantages of one vs the other, and so on.

Advance: the study adds to the current literature as the study of tumor hypoxia in 3D remains a complicated issue. The interesting co-culture settings with macrophages suggests potential uses of this model to study cell-cell interactions.

Audience: the study is very methodological and offers a tool that could be used by cancer biologists - and maybe by other biology fields.

Reviewer #2

Evidence, reproducibility and clarity:

Summary

Anandi and colleagues present a manuscript describing a nice assay for exploring the progressive effect of metabolic depletion of the nutrients and oxygen on the invasion of cancer cells. This builds upon and extends a device that they previously described - MEMIC - and now enables 3D analysis of small numbers of cells. The key to their method is the inclusion of a layer of consumer cells that deplete oxygen and nutrients. Using this tool, they demonstrate that depleted environments promote invasive behavior and lower cell-cell adhesion. This is related to the nutrient-deprived and hypoxic environments found in the center of many tumors. Cellular Potts Modelling is used to explore ideas around the cooperation between reduced cell-cell adhesion and increase ECM adhesion in promoting invasion. Overall, this is a well-constructed manuscript that will be of interest to cell biologists and cancer biologists.

Major comments

I realize this work is submitted to review commons and this complicates the recommendation regarding publication. My view is that the 'more prestigious' journals would require greater mechanistic insight, but that the work could find a suitable place in other members of the review commons stable. My comments are divided into those essential for any journal and those that might be journal dependent.

We hope that the mechanistic experiments added to our new manuscript version will appeal the reviewer and merit publication in any of the review commons journals.

Essential regardless of journal

1. Many of the figures lack information about the number of spheroids analyzed and from how many biological repeats they are derived.

We have now added this information to all our experiments. This information can be found in the figures and on the figure legends.

2. The authors need to provide citations for their assertion that only gases can cross the PDMS, but not other small metabolites. They should also comment on whether the build-up of CO₂ might be relevant.

We have now added the original reference where they describe PDMS's properties (Cox and Dunn, 1986).

The point raised about CO₂ is very interesting, but we do not expect a buildup of this gas. When using PDMS, CO₂ would not accumulate as PDMS membranes are permeable to gases – including CO₂. When using glass covers, the lack of oxygen

should minimize CO₂ production as hypoxic cells will not be able to conduct oxidative phosphorylation and produce lactic acid instead.

3. *The data on the directionality of migration when consumers are present are not significant and doesn't warrant the speculation in lines 186-189.*

Following the reviewer's advice we have removed this speculation.

4. *The ECM degradation in Figure 3 should be quantified.*

We agree. We added additional quantifications for the gelatin degradation assay. We also highlight the quantification we already had of the ECM degradation assessed via DQ collagen. Those data can be found in the new figures 4 and S4, respectively.

5. *Do the authors have evidence that the hypoxia-exposed cells are more adhesive to ECM. This is central to their Potts model and I could not locate the supporting experimental data. If not, then the Potts model should include matrix proteolysis, which they do have data about.*

Again, this is a very insightful observation, and we completely understand this confusion. We think that this may part of the inherent challenge of trying to condense biological problems into analogies or "metaphors" when using physical/mathematical models.

The algorithm in a Cellular Potts model (CPM) tries to minimize the energy of the system (the entire group of cells/ECM that we are modelling). This global energy reduction is achieved by minimizing local energies in the cell-cell and cell-ECM interactions. The way the algorithm executes this minimization, is by always (probability $p=1$) accepting a configuration that decrease the energy while restricting the configurations that lead to higher energies (with a probability of $p = e^{-\Delta H/T}$) where ΔH is the difference between the current and previous energy.

So, the only thing the model is really doing is to increase the likelihood that cells are in a more "comfortable" environment – *i.e.* that the energy from the interactions with their neighboring cells and ECM is as low as possible. For example, if cell 1 and cell 2 adhere strongly but not to cell 3, in a CPM this is modelled as a low ΔH between cell 1 and 2 and a higher ΔH with cell 3. Conversely, when people model cells better at "invading" into a new "territory" they choose a lower energy between that cell type and that type of substratum.

In other words, our CPM does not "care" whether ischemic cells invade the ECM because they create space through increased proteolysis or because they are more adherent to the ECM. These two scenarios are the same in a CPM and it is consistent with previous CPM models of similar scenarios (e.g.: PMID: 18835895, 33933478, 26436883, 23596570).

We have now reworded the description of the model on the main text, and we added an illustration hoping to make this aspect of the model clearer (Fig. S4F).

6. Is the down-regulation of E-cadherin transcriptional - i.e. is the mRNA level reduced?

This is a great question. After the reviewer posed this question, we looked at our data and we concluded E-cad's downregulation is transcriptional. Assessing local mRNA levels in the 3MIC is challenging. However, our E-Cad reporter (pHAGE-E-cadherin-RFP, addgene #79603) is a red fluorescent protein driven by the CDH1 (E-Cad) promoter. RFP levels decrease with ischemia indicating that this regulation occurs at the promoter/transcriptional level. We now added this point to the revised manuscript (lines 259-261). We thank the reviewer for this insight!

7. The title of figure 6 is misleading. The authors do not demonstrate chemoresistance in terms of cell survival or cell proliferation, which is how the term is normally used. The authors should measure cell number, proliferation, and cell viability. The data presented in the Supplementary Figure are inadequate with no quantification. The FUCCI reporter cells would be a good tool for this. Also, why use 150nM paclitaxel when the IC50 is 817nM? This seems bizarre. Lastly, there is a typo in the figure that suggest 150mM drug was used.

We apologize if these experiments caused confusion. Our intention was to look at the anti-migratory effects of Taxol-related drugs. As such, we first determined the concentrations at which the drug was lethal to our cells (this is the LD50 of ~800nM). Then, we tested if lower concentrations – which we knew were not lethal – would affect cell migration, protrusions, etc. Hence the 30-150nM range we used in our experiments.

We have now completely rewritten this section hoping that our approach is now clearer. We have also changed the title of the section and the figure legend to clarify that we are studying the effects of Taxol as anti-motility drug rather than its effects on cell survival and proliferation (now Fig. 7). Finally, we have now fixed the 150mM/150nM typo in the figure legend.

Journal dependent

1. The authors have not excluded that either changes in nutrients, or even a pro-invasive factor, produced by consumer cells are necessary for the increased invasion. They have only shown that they are not sufficient. The authors should perform a series of experiments comparing hypoxic conditions with normal media and normoxic conditions with nutrient depleted/condition media by prior culturing of KP cancer cells.

This is a great point. We actually do not want or propose to exclude this possibility. So, we have now added text to clarify this issue (lines 431-435).

In fact, we would be thrilled if there is a pro-invasive factor. If that would be the case, our results indicate this factor is only effective under ischemia. Because the same consumer cells do not have an effect on the same type of tumor spheroids under well-nurtured environments. In addition, our new pH measurements and perturbations experiments agree with this reviewer's intuition about additional

factors being key in the increased invasion (see new Figure 2). We are very excited about these new results, and we hope this reviewer will be excited too.

2. What is the oxygen sensor for increased invasion? PHD1-3 would be a good place to start looking. Is the PHD2-HIF axis important? Do VHL mutant cells still show responses to the consumer cells?

Following the reviewer's feedback, we generated isogenic HIF1A KO cell lines to study whether HIF1A was directly needed in the invasion of tumor spheroids within the 3MIC. We complemented these loss-of-function experiments with For HIF1A gain-of-function using pharmacological interventions that stabilize HIF1A under normal oxygen levels (CoCl₂ and DMOG).

As shown in the new figure 2, these experiments mirrored our hypoxia experiments: HIF1A activity was not sufficient but it was required to drive the invasion of ischemic spheroids. We think that these new results are particularly interesting when taken together with our new pH-perturbation experiments. Briefly, our new experiments results show that in addition to the requirement of hypoxia/HIF1A, media acidity also has a strong effect on spheroid invasion. More excitingly, a drop in pH is sufficient to dramatically increase invasion – even in control well-nurtured spheroids. We think that the effects of pH and hypoxia are linked. HIF1A activation and hypoxia the increase glycolysis and thus lactic acid secretion. We speculate that this glycolytic switch is where hypoxia is important, but it is not sufficient because under well-perfused conditions (e.g. healthy tissue or large culture media volume) lactic acid levels may not buildup enough to significantly lower the extracellular pH. In contrast, under poor perfused conditions (3MIC and solid tumors) or if we flood cell cultures with lactic acid, the media's pH drops dramatically (Fig. 2).

3. If they include both spheroids of endothelial cells and cancer cells, will the resulting protrusions in hypoxia grow towards each other? Would macrophages enhance this process?

We agree with the reviewer this is an interesting question and we have anecdotally observed this effect. In the manuscript, we used these chimeric endothelial/tumor spheroids rather than separate ones (Fig. 6E). We do not find strong evidence that their protrusions grew towards each other, but this is something that we would like to explore in the future with more detail.

Significance:

The main advance is technical, as many previous studies have related hypoxia to increased cancer cell invasion, which the authors correctly acknowledge and cite. It is scholarly study, which will be of interest to many readers, and the method reported is likely to be adopted by several groups.

Reviewer #3

Evidence, reproducibility and clarity:

In this work, Anandi et al., developed a cell culture system to live image the initial transformation of cells in deprivation of oxygen and nutrients in a 3D context. Using this system, 3MIC, they were able to create oxygen and nutrient gradients to simulate ischemic conditions that arise deep within tumors and that typically precede metastasis. With the 3MIC system they validated that ischemia triggers cell migration and invasion of tumor cells. In addition, 3MIC also allowed them to study the interaction of tumor spheroids with stromal cells such as macrophages and endothelial cells. Interestingly, the authors showed that co-culturing tumor spheroids with stromal cells increased the pro-metastatic features induced by ischemia conditions. Lastly, using 3MIC allowed the authors to discern that a poor paclitaxel response in ischemic-like cells is driven by intrinsic cellular resistance rather than due to lower drug concentration.

Overall, the work is very well-written, and the results are consisting, convincing and support the conclusions. The methods are clear and complete and allow the reproducibility of the experiments. The experiments are adequately replicated and statistical analyses are well described. However, I have few suggestions to improve the impact of the manuscript:

1. The authors conclude that 3MIC results in the accumulation of lactic acid and nutrient deprivation in an increasing manner when moving far from the opening site. Is there a way to actually show this? So far, the authors employ a hypoxia sensor only. A sensor for internal pH or other method for nutrient deprivation would help to support the conclusion and further validate the model.

This is an excellent point. Following the reviewer's feedback, we tested additional sensors including for extra- and intra-cellular pH. As mentioned above, we observed dramatic changes in extracellular pH levels. We followed up these observations with a series of experiments that showed a key functional role for media acidification in driving invasion (Figure 2).

2. According to figure S3E, the main cell line used by the authors is already quite mesenchymal. It would be good to know if the results showed here are consistent in cells with a more basal epithelial phenotype. Do epithelial cells need stronger ischemic conditions to undergo phenotypic changes?

This is a great catch. To explore this further, we run a Western Blot analysis to compare epithelial and mesenchymal markers expressed by the main cells we used here (Lung KPs) and to compare them to levels in a stereotypical epithelial (MCF-7) and a mesenchymal (MDA-MB-231) cell line (new Fig. S4D). As the reviewer correctly points out, we do see that E-Cad and Vimentin are co-expressed in KP cells.

So far, our observations in a range of cell lines are a consistent decrease in E-Cad levels with no significant effects in vimentin levels – regardless of the basal levels of this protein.

Interestingly, a recent study¹ demonstrated in triple-negative breast cancer models, that an EMT hybrid phenotype – including the presence of Vimentin – is required for metastasis. A compelling hypothesis then is that ischemia in the tumor microenvironment may favor these hybrid phenotypes. We briefly discuss this topic in the revised version of this manuscript.

3. *The number of replicates should be included in each figure legend and not only in the methods section. From data presented it is not clearly stated what do points mean in boxplots (e.g, Fig1H, 2B,G...). How many cells/spheroids did the authors count in each experiment?*

We have now added this information to all our experiments. This information can be found in the figures and on the figure legends.

4. *Figure 3B is not mentioned in the main text.*

We apologize for this error, and we thank the reviewer for catching this issue, which have now corrected.

5. *Line 295: "In the absence of macrophages, clusters of endothelial cells remained mostly rounded, even in the presence of consumer cells and regardless of their location along the ischemic gradient (Fig. 5A; Video S6)." However, in Video S6, both images show endothelial cells co-cultured with macrophages. I consider that Video S6 should be not referenced here.*

The reviewer is correct so have removed that reference.

6. *References style should be homogeneous (e.g, in Ref 13 appears "Nature Reviews Cancer" whereas in Ref 14 "Nat Rev Cancer"). Also, in Ref 25, the journal is missing.*

We apologize for this oversight, and we have not tried to be more consistent in our references.

7. *In plots where distance to open chamber site is not especificy (e.g. 6B), at what distance were the data recorded? Please, indicate in the figure legend.*

We have now added this information to our figures.

8. *In the experiment showed in Fig 4, the sorting strategy would include stromal cells such as fibroblasts and endothelial cells in the GFP- population (as only CD45+ cells are removed). These cells will likely also grow in the 3MIC system and have an effect in migration. Can the authors rule out this confounding effect?*

¹ Grasset et al. *Triple-negative breast cancer metastasis involves complex epithelial-mesenchymal transition dynamics and requires vimentin*. Science Transl Med. 2022 4(656) PMID: 35921474.

The reviewer is correct. We still think that the possibility of fibroblast contamination is low. First, the fluorescence of HRE-GFP cells under normoxic, is still higher than the autofluorescence of cells not expressing this constructs (such as fibroblasts). This is quite normal as most sensors/reporter have some leakage and thus there is a small amount of transcription. Second, intradermal and subcutaneous tumors are quite poor in fibroblasts. In fact, to study the role of fibroblasts in these tumors, they are usually co-injected with tumor cells (PMID: 20138012). Third, in the process of tumor dissociation and *in vitro* establishment, non-transformed cells tend to die more. Since these are more technical points, we moved the cell sorting details to the material and methods section.

9. In Fig 5C the panel of proximal + macrophages is missing

We apologize for this mistake, and we have corrected in the new version of the manuscript.

10. In Fig. 5, Linifanib is used to study the effect of blocking VEGF. Linifanib can also interact with RTKs and PDGF. This fact should be acknowledged.

We agree with this point. Following the reviewer's advice, we now acknowledged the potential off-target effects of these inhibitors (lines 354-355).

Significance

This is a very interesting work with the development of a simple and cost-effective system that allows to continuously monitor biological processes in 3D cultures under nutrient-modified conditions. In general, these data would be broadly interesting to cancer community in general, as 3MIC is a very versatile system, where several aspects can be studied and precisely discerned.

Re: Life Science Alliance manuscript #LSA-2024-03053-T

Carlos Carmona-Fontaine
New York University

Dear Dr. Carmona-Fontaine,

Thank you for submitting your manuscript entitled "Direct visualization of emergent metastatic features within a model of the tumor microenvironment" to Life Science Alliance. We invite you to submit a revised manuscript addressing Reviewer 2's comments about statistics.

Thank you for this interesting contribution to Life Science Alliance. We are looking forward to receiving your revised manuscript.

Sincerely,

- A letter addressing the reviewers' comments point by point.
- An editable version of the final text (.DOC or .DOCX) is needed for copyediting (no PDFs).
- High-resolution figure, supplementary figure and video files uploaded as individual files: See our detailed guidelines for preparing your production-ready images, <https://www.life-science-alliance.org/authors>
- Summary blurb (enter in submission system): A short text summarizing in a single sentence the study (max. 200 characters including spaces). This text is used in conjunction with the titles of papers, hence should be informative and complementary to the title and running title. It should describe the context and significance of the findings for a general readership; it should be written in the present tense and refer to the work in the third person. Author names should not be mentioned.

B. MANUSCRIPT ORGANIZATION AND FORMATTING:

*****IMPORTANT:** It is Life Science Alliance policy that if requested, original data images must be made available. Failure to provide original images upon request will result in unavoidable delays in publication. Please ensure that you have access to all original microscopy and blot data images before submitting your revision. *******

=====

RE: Life Science Alliance Manuscript #LSA-2024-03053-TR

Dr. Carlos Carmona-Fontaine
New York University
100 Washington Place
NYC 10003

Dear Dr. Carmona-Fontaine,

Thank you for submitting your revised manuscript entitled "Direct visualization of emergent metastatic features within a model of the tumor microenvironment". We would be happy to publish your paper in Life Science Alliance pending final revisions necessary to meet our formatting guidelines.

- please be sure that the authorship listing and order is correct
- please use the [10 author names, et al.] format in your references (i.e. limit the author names to the first 10)
- please consult our manuscript preparation guidelines <https://www.life-science-alliance.org/manuscript-prep> and make sure your manuscript sections are in the correct order
- please add a separate figure legend section (including your main, supplementary and video legends) to the main manuscript text
- please add the panels for Figure I and J to the figure legend for Figure S3

A. FINAL FILES:

B. MANUSCRIPT ORGANIZATION AND FORMATTING:

Sincerely,

3rd Editorial Decision

RE: Life Science Alliance Manuscript #LSA-2024-03053-TRR

Dr. Carlos Carmona-Fontaine
New York University
100 Washington Place
NYC 10003

Dear Dr. Carmona-Fontaine,

Thank you for submitting your Methods entitled "Direct visualization of emergent metastatic features within a model of the tumor microenvironment". It is a pleasure to let you know that your manuscript is now accepted for publication in Life Science Alliance. Congratulations on this interesting work.

DISTRIBUTION OF MATERIALS:

Again, congratulations on a very nice paper. I hope you found the review process to be constructive and are pleased with how the manuscript was handled editorially. We look forward to future exciting submissions from your lab.

Sincerely,
